# Partially Equivariant Reinforcement Learning in Symmetry-Breaking Environments

**Junwoo Chang**[1], **Minwoo Park**[2], **Joohwan Seo**[3], **Roberto Horowitz**[3], **Jongmin Lee**[2*],
**Jongeun Choi**[1,2*]
[1]School of Mechanical Engineering, Yonsei University, Seoul, South Korea
[2]Department of Artificial Intelligence, Yonsei University, Seoul, South Korea
[3]Department of Mechanical Engineering, University of California, Berkeley, United States
{junwoochang, minwoopark, jongminlee, jongeunchoi}@yonsei.ac.kr
{joohwan_seo, horowitz}@berkeley.edu

## Abstract

Group symmetries provide a powerful inductive bias for reinforcement learning (RL), enabling efficient generalization across symmetric states and actions via group-invariant Markov Decision Processes (MDPs). However, real-world environments almost never realize fully group-invariant MDPs; dynamics, actuation limits, and reward design usually break symmetries, often only locally. Under group-invariant Bellman backups for such cases, local symmetry-breaking introduces errors that propagate across the entire state–action space, resulting in global value estimation errors. To address this, we introduce Partially Group-Invariant MDP (PI-MDP), which selectively applies group-invariant or standard Bellman backups depending on where symmetry holds. This framework mitigates error propagation from locally broken symmetries while maintaining the benefits of equivariance, thereby enhancing sample efficiency and generalizability. Building on this framework, we present practical RL algorithms – Partially Equivariant (PE)-DQN for discrete control and PE-SAC for continuous control – that combine the benefits of equivariance with robustness to symmetry-breaking. Experiments across Grid-World, locomotion, and manipulation benchmarks demonstrate that PE-DQN and PE-SAC significantly outperform baselines, highlighting the importance of selective symmetry exploitation for robust and sample-efficient RL. Project page: https://pranaboy72.github.io/perl_page/

## 1 Introduction

Group symmetries provide a powerful inductive bias in machine learning, enabling models to generalize efficiently. In robotics and continuous control, leveraging equivariance has been shown to improve data efficiency in both behavior cloning (Zeng et al., 2021; Ryu et al., 2023; 2024; Wang et al., 2024; Tie et al., 2025; Huang et al., 2024; Zhao et al., 2025; Seo et al., 2024; 2023; 2025a;b), where the data collection is costly, and reinforcement learning (RL) (Van der Pol et al., 2020; Kohler et al., 2024; Wang et al., 2022a;b; Tangri et al., 2024; Nguyen et al., 2023; Finzi et al., 2021a; Park et al., 2025), where exploration can be inefficient. Most existing equivariant RL methods are grounded in the notion of a group-invariant Markov Decision Process (MDP) (Wang et al., 2022b; 2021), where invariance of the reward and transition functions implies symmetry in the optimal policy.

In practice, however, these symmetry assumptions rarely hold exactly. Real-world environments introduce *symmetry-breaking factors* such as dynamics, actuation limits, or reward shaping. Under the Bellman backups based on the group-invariant MDP, even local violations of symmetry can introduce errors that propagate across the state–action space, leading to degraded value estimates, suboptimal policies, or even training failure. Prior works on approximate equivariance (Finzi et al., 2021a; Park et al., 2025) attempt to mitigate this challenge by relaxing equivariance globally, e.g., by modifying architectures to tolerate violations. While effective to some extent, these methods often

---
*Equal advising

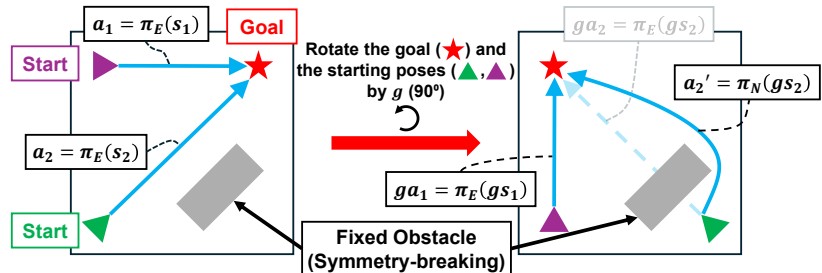

Figure 1: **Overview of Partially Equivariant RL.** Assume a simple goal-reaching task where the state space includes only agent and goal poses. When states are rotated by 90°, the purple agent's equivariant action $ga_1$ remains valid. Conversely, the green agent's rotated action $ga_2$ collides with the obstacle, illustrating local symmetry-breaking in the true dynamics and restricting the use of strict rotational symmetry. To address this, our framework learns a gating function $\lambda$ that detects such violations and activates an unconstrained policy $\pi_N$ for $a_2'$, retaining the sample efficiency of equivariance while falling back to $\pi_N$ only where group symmetry is invalid.

lose the sample efficiency benefits of strict equivariance and can become unstable when symmetry-breaking is extensive, since equivariance is still applied indiscriminately across the entire space.

To overcome this limitation, we introduce the framework of the **Partially Group-Invariant MDP (PI-MDP)**, which selectively applies the group-invariant structure only in regions where symmetry is preserved (Fig. 1). Our approach builds on the derivation that local symmetry-breaking leads to one-step backup errors that propagate globally. By routing updates to standard Bellman updates under the true MDP, we limit the propagation of one-step backup errors across the space. In particular, we detect symmetry-breaking regions via predictor disagreement outliers between an equivariant and an unconstrained one-step predictor, and apply standard rather than equivariant updates on those outliers while retaining equivariance elsewhere. Building on this framework, we develop practical reinforcement learning algorithms for both discrete and continuous control that retain the benefits of equivariance in symmetric regions while remaining robust to substantial symmetry-breaking.

The contributions of our work are summarized as follows: 1) We analyze how local symmetry violations induce global value error via one-step backups, clarifying when selective symmetry is beneficial. 2) We introduce the Partially Group-Invariant MDP (PI-MDP) and a practical RL formulation that uses equivariance where symmetry holds and falls back to standard updates where it breaks. 3) Across state-based discrete and continuous control experiments, we show that our method retains the sample efficiency gains of equivariance in symmetric regions and remains robust as symmetry-breaking increases, outperforming strict and approximate-equivariant baselines.

## 2 RELATED WORK

**Group equivariance in continuous control.** Recent works have applied group equivariance to imitation learning and classical control (Zeng et al., 2021; Ryu et al., 2023; 2024; Wang et al., 2024; Tie et al., 2025; Huang et al., 2024; Zhao et al., 2025; Seo et al., 2024; 2023; 2025a), demonstrating high data efficiency and generalization over baseline models. Parallel efforts have investigated group equivariance in reinforcement learning (RL) (Van der Pol et al., 2020; Kohler et al., 2024; Wang et al., 2022a;b; Tangri et al., 2024; Nguyen et al., 2023), showing improved sample efficiency compared to the conventional RL approaches. However, the effectiveness of equivariant RL remains limited in more general settings, such as robotic control tasks, where inherent symmetry-breaking often arises from factors including occlusions, environmental asymmetries, kinematic singularities, and complex dynamics.

**Approximate equivariance.** Recent studies have proposed relaxing strict group equivariance to handle symmetry-breaking in data (Finzi et al., 2021a; Park et al., 2025; Wang et al., 2022c; Romero & Lohit, 2022; van der Ouderaa et al., 2022; Hofgard et al., 2024). Such approaches introduce approximate equivariance, enabling models to remain robust when exact symmetries do not hold. In

reinforcement learning, approximate equivariant architectures have also shown improved robustness and efficiency against symmetry-breaking (Finzi et al., 2021a; Park et al., 2025). However, these methods primarily focus on global relaxations of equivariance at the representation level. In contrast, our approach addresses symmetry-breaking by minimizing local equivariance errors during the Bellman backup, thereby preventing their global propagation through value updates.

## 3 PRELIMINARIES

**Reinforcement learning.** We consider a Markov decision process (MDP) defined as $\mathcal{M} = (\mathcal{S}, \mathcal{A}, P, R, \gamma)$ where $\mathcal{S}$ is the state space, $\mathcal{A}$ is the action space, $R : \mathcal{S} \times \mathcal{A} \to \mathbb{R}$ is the reward function, $P(\cdot \mid s, a)$ is the transition kernel, and $\gamma \in (0, 1)$ is the discount factor. The agent learns a policy $\pi$ to maximize the expected return, $J = \mathbb{E}_{\pi, P}\left[ \sum_{t=0}^{\infty} \gamma^t r_t \,\middle|\, s_0 = s, a_0 = a \right]$. The Bellman operator under a policy $\pi$ is $(\mathcal{T}^\pi Q)(s, a) = R(s, a) + \gamma \, \mathbb{E}_{s' \sim P(\cdot|s,a)}\left[ \mathbb{E}_{a' \sim \pi(\cdot|s')}[Q(s', a')] \right]$, while the optimal (hard) Bellman operator is $(\mathcal{T}Q)(s, a) = R(s, a) + \gamma \, \mathbb{E}_{s' \sim P(\cdot|s,a)}\left[ \max_{a'} Q(s', a') \right]$.

**Group equivariance.** A *symmetry* is a transformation that preserves certain properties of a system, and the set of all symmetries forms a **group** (Bronstein et al., 2021). A **group representation** is a homomorphism $\rho : G \to GL(n)$ that maps each group element $g \in G$ to an invertible $n \times n$ matrix. A function $f : X \to Y$ is **equivariant** if $\rho_Y(g)f(x) = f(\rho_X(g)x)$, and **group-invariant** if $f(x) = f(\rho_X(g)x)$, $\forall g \in G, x \in X$.

**Group-invariant MDP.** A group-invariant MDP (Wang et al., 2022b; 2021) is an abstract MDP based on MDP homomorphisms (Ravindran & Barto, 2001; 2004), denoted as $\mathcal{M}_G = (\mathcal{S}, \mathcal{A}, P, R, \gamma)$. The optimal policy and optimal $Q$-function of the original MDP are recoverable from the abstract MDP provided the reward and transition kernel are group-invariant:

$$R(s, a) = R(gs, ga), \quad P(s' \mid s, a) = P(gs' \mid gs, ga), \quad \forall g \in G.$$

## 4 SYMMETRY-BREAKING IN GROUP-INVARIANT MDPS

Most equivariant RL approaches assume the existence of a group-invariant MDP (Sec. 3) (Wang et al., 2021; 2022b; Mondal et al., 2022; Van der Pol et al., 2020; Tangri et al., 2024). However, many continuous control tasks (e.g., robotics) violate these assumptions in certain regions of the state–action space. We begin by analyzing how such **symmetry-breaking** perturbs Bellman backups and subsequently propagates into the learned value function.

Let $\mathcal{M}_N(\mathcal{S}, \mathcal{A}, R_N, P_N, \gamma)$ denote the standard environment, and let $\mathcal{M}_E(\mathcal{S}, \mathcal{A}, R_E, P_E, \gamma)$ be a group-invariant MDP defined on the same spaces. To construct such a group invariant MDP from $\mathcal{M}_N$, we average the original rewards and dynamics over the symmetry group $G$:

$$R_E(s, a) = \int_G R_N(gs, ga) \, d\mu(g) \quad P_E(s'|s, a) = \int_G P_N(gs'|gs, ga) \, d\mu(g),$$

where $d\mu(g)$ is the normalized Haar measure on $G$ (uniform measure for finite groups). This averaging ensures that $R_E$ and $P_E$ satisfy the group-invariance condition, thereby making $\mathcal{M}_E$ the canonical group-invariant approximation of $\mathcal{M}_N$. For $(s, a) \in \mathcal{S} \times \mathcal{A}$, define pointwise discrepancies between $\mathcal{M}_N$ and $\mathcal{M}_E$ via

$$\epsilon_R(s, a) := |R_N(s, a) - R_E(s, a)|,$$
$$\epsilon_P(s, a) := \tfrac{1}{2} \int_{\mathcal{S}} \big| P_N(s' \mid s, a) - P_E(s' \mid s, a) \big| \, ds', \tag{1}$$

where $\epsilon_R$ is the absolute reward difference and $\epsilon_P$ is the total-variation distance between next-state kernels. Using these pointwise discrepancies, we formalize per-state–action symmetry-breaking as follows.

**Definition 1** (Per-state–action symmetry-breaking). *Consider the true MDP $\mathcal{M}_N$. We say that symmetry is **preserved** at $(s, a)$ if both $\epsilon_R(s, a) = 0$ and $\epsilon_P(s, a) = 0$; in this case, the group-invariant approximation $\mathcal{M}_E$ locally coincides with $\mathcal{M}_N$. Conversely, if $\epsilon_R(s, a) > 0$ or $\epsilon_P(s, a) > 0$, we say that symmetry is **broken** at $(s, a)$.*

In the terminology of Wang et al. (2023), state–action pairs with $\epsilon_R(s, a) > 0$ or $\epsilon_P(s, a) > 0$ fall into symmetry-breaking regimes (e.g., incorrect equivariance), where the assumed group action does not match the true MDP dynamics. Our formalism captures this mismatch at the MDP level via the reward and transition discrepancies $(\epsilon_R, \epsilon_P)$ and the resulting Bellman error in Lemma 1.

We now quantify how these local mismatches perturb a single Bellman backup and how this error propagates to the optimal value functions. Let $\mathcal{T}_i$ denote the Bellman optimality operator in MDP $i \in \{N, E\}$. We assume rewards are uniformly bounded as $|R_i(s, a)| \leq R_{\max}$ and that the discount factor satisfies $\gamma \in (0, 1)$; define $V_{\max} := R_{\max}/(1 - \gamma)$.

**Lemma 1** (One-step Bellman error). *For any bounded $Q$ and any $(s, a) \in \mathcal{S} \times \mathcal{A}$,*

$$\left| (\mathcal{T}_N Q)(s, a) - (\mathcal{T}_E Q)(s, a) \right| \leq \epsilon_R(s, a) + 2\gamma \|V_Q\|_\infty \epsilon_P(s, a).$$

Here $V_Q(s') = \max_{a'} Q(s', a')$ and $\|V_Q\|_\infty \leq \|Q\|_\infty$. If $Q$ is an action–value function, then $\|Q\|_\infty \leq V_{\max}$, hence $\|V_Q\|_\infty \leq V_{\max}$ and we define the pointwise bound

$$\delta(s, a) := \epsilon_R(s, a) + 2\gamma V_{\max} \epsilon_P(s, a). \tag{2}$$

We next show that this local error lifts to a global bound on the optimal action–value functions via contraction.

**Proposition 1** (Value-function gap). *Let $Q_i^*$ be the optimal action–value function in MDP $i$. Then*

$$\|Q_N^* - Q_E^*\|_\infty \leq \frac{1}{1 - \gamma} \|\delta\|_\infty.$$

The proofs of Lemma 1 and Proposition 1 are provided in Appendix A.1

*Intuition.* Local symmetry-breaking introduces a one-step Bellman backup error $\delta(s, a)$ which propagates through repeated backups and is amplified by the factor $(1 - \gamma)^{-1}$ due to contraction. This results in a global deviation bounded by $\frac{1}{1-\gamma} \|\delta\|_\infty$, which can cause suboptimal policies or unstable training. We visualize this propagation in a Grid-World example, and show that a strictly equivariant policy can fail to learn (Appendix E). Prior works mitigate such errors with global relaxations (Finzi et al., 2021a; Park et al., 2025), whereas our approach employs **local** corrections that are less conservative and effective when symmetry holds only piecewise.

## 5 PARTIAL GROUP-INVARIANCE IN MARKOV DECISION PROCESSES

In what follows, we present an efficient method for handling local symmetry-breaking. Specifically, we propose a **Partially Group-Invariant MDP (PI-MDP)** that interpolates, for each state–action pair, between a group-invariant MDP and the true environment.

### 5.1 PARTIALLY GROUP-INVARIANT MDP

**Definition 2** (PI-MDP). *Let the true MDP be $\mathcal{M}_N = (\mathcal{S}, \mathcal{A}, R_N, P_N, \gamma)$ and the group-invariant MDP be $\mathcal{M}_E = (\mathcal{S}, \mathcal{A}, R_E, P_E, \gamma)$, sharing the same $(\mathcal{S}, \mathcal{A}, \gamma)$. Define a Partially Group-Invariant MDP (PI-MDP) $\mathcal{M}_H = (\mathcal{S}, \mathcal{A}, R_H, P_H, \lambda, \gamma)$ with a measurable gating function $\lambda : \mathcal{S} \times \mathcal{A} \to [0, 1]$,*

$$R_H(s, a) := (1 - \lambda(s, a)) R_E(s, a) + \lambda(s, a) R_N(s, a),$$
$$P_H(\cdot \mid s, a) := (1 - \lambda(s, a)) P_E(\cdot \mid s, a) + \lambda(s, a) P_N(\cdot \mid s, a).$$

Since $0 \leq \lambda(s, a) \leq 1$ for all $(s, a)$ and both $(R_E, P_E)$ and $(R_N, P_N)$ are valid, $(R_H, P_H)$ defines a valid MDP. We then characterize the partially group-invariant optimality operator induced by the gating function.

**Theorem 1** (Partially group-invariant optimality operator). *Let $\mathcal{T}_i$ denote the (hard) Bellman optimality operator in MDP $i \in \{E, N, H\}$, $(\mathcal{T}_i Q)(s, a) = R_i(s, a) + \gamma \mathbb{E}_{s' \sim P_i(\cdot|s, a)}[\max_{a'} Q(s', a')]$. For any bounded $Q : \mathcal{S} \times \mathcal{A} \to \mathbb{R}$ and all $(s, a)$,*

$$(\mathcal{T}_H Q)(s, a) = (1 - \lambda(s, a)) (\mathcal{T}_E Q)(s, a) + \lambda(s, a) (\mathcal{T}_N Q)(s, a). \tag{3}$$

*If $|R_E|, |R_N| \leq R_{\max}$ and $\gamma \in (0, 1)$, then $\mathcal{T}_H$ is a $\gamma$-contraction and has a unique fixed point $Q_H^*$.*

We next bound the deviation of the fixed point from the true optimum.

**Corollary 1** (Proximity bound). *Let $Q_N^*$ be the optimal action–value of the true MDP $\mathcal{M}_N$, and let $\delta(s,a)$ be the one-step pointwise Bellman error bound. Then*

$$\|Q_H^* - Q_N^*\|_\infty \ \leq \ \frac{1}{1-\gamma} \left\| (1-\lambda)\,\delta \right\|_\infty. \tag{4}$$

Moreover, the right-hand side of Eq. (4) is zero whenever, at every $(s,a)$, either $\lambda(s,a) = 1$ (the gating function routes to the true MDP) or the group-invariant MDP coincides with the true MDP at $(s,a)$, that is, $R_E(s,a) = R_N(s,a)$ and $P_E(\cdot \mid s,a) = P_N(\cdot \mid s,a)$. Consequently, symmetric pairs contribute zero via MDP coincidence, and symmetry-breaking pairs contribute zero when $\lambda$ correctly gates to 1. The proofs of Theorem 1 and Corollary 1 can be found in Appendix A.2.

*Intuition.* By gating the reward and transition kernels, the PI-MDP is itself a valid MDP. Its optimality operator satisfies the affinity identity in Eq. (3). Since a convex combination of $\gamma$-contraction is again a $\gamma$-contraction, $\mathcal{T}_H$ admits a unique fixed point $Q_H^*$. Corollary 1 bounds the deviation from the true optimum: the gap $\|Q_H^* - Q_N^*\|_\infty$ is controlled by the gated mismatch term on the right-hand side of Eq. (4), scaled by $(1-\gamma)^{-1}$. When $\lambda$ correctly localizes symmetry-breaking, $Q_H^*$ closely tracks $Q_N^*$ while reverting to the group-invariant MDP where symmetry holds. We provide the extension of the PI-MDP to the entropy-regularized (soft) setting in Appendix A.3.

**Remark 1** (Hard gating). *All results above hold for any measurable gating function $\lambda : \mathcal{S} \times \mathcal{A} \to [0,1]$. When $\lambda(s,a) \in \{0,1\}$, the PI-MDP routes pointwise to $(R_E, P_E)$ on symmetric pairs and $(R_N, P_N)$ otherwise. In our algorithms, we adopt this hard-gating regime for simplicity and empirically more stable training (Fig. 7).*

## 6 PARTIALLY EQUIVARIANT REINFORCEMENT LEARNING

This section introduces partially equivariant reinforcement learning (Algorithm 1) for the PI-MDP setting (Sec. 5.1). We (i) learn a gating function $\lambda(s,a)$ that localizes symmetry-breaking, and (ii) couple $\lambda$ to equivariant and unconstrained value/policy heads.

### 6.1 LEARNING $\lambda(s,a)$ VIA DISAGREEMENT SUPERVISION

By Corollary 1, the value gap vanishes when $\lambda(s,a) = 1$ on symmetry-breaking pairs and $\lambda(s,a) = 0$ where the proxy and true MDPs coincide (assuming an oracle binary gate under local symmetry). To approximate this behavior, we train a gating function $\lambda_\omega(s,a) \in \{0,1\}$ using the *disagreement* between two one-step predictors: an equivariant predictor $\hat{P}_E : \mathcal{S} \times \mathcal{A} \to \mathbb{R}^n$ constrained to respect the group symmetries of $\mathcal{M}_E$ and an unconstrained predictor $\hat{P}_N : \mathcal{S} \times \mathcal{A} \to \mathbb{R}^n$, where $n$ is the dimension of the predictor output.

Concretely, $\hat{P}_E$ and $\hat{P}_N$ are trained on transitions $(s,a,r,s')$ to approximate the one-step MDP components, i.e., the transition kernel and, when used, the reward function. We define a scalar disagreement score

$$d(s,a) = D\big(\hat{P}_E(\cdot \mid s,a), \ \hat{P}_N(\cdot \mid s,a)\big),$$

where $D$ is a discrepancy measure (e.g., squared error between next-state predictions, total-variation distance between transition distributions, or an $\ell_1$ distance between predicted rewards). Detailed implementations of $\hat{P}_E$, $\hat{P}_N$, and $D$ are given in Appendix B.1.

At symmetric pairs (i.e., where $\epsilon_R(s,a) = \epsilon_P(s,a) = 0$), the equivariant predictor $\hat{P}_E$ is consistent with $\mathcal{M}_E$, which coincides with the true MDP $\mathcal{M}_N$, and the unconstrained predictor $\hat{P}_N$ can match the same behavior. In this case, the disagreement $d(s,a)$ remains small. At symmetry-breaking pairs, the assumption of group-invariance is violated: by construction, $\hat{P}_E$ can only represent the group-averaged surrogate $P_E$, whereas $\hat{P}_N$ can approximate the true kernel $P_N$, so their predictions diverge. Consequently, $d(s,a)$ tends to be larger precisely in those regions where $(R_E, R_N)$ or $(P_E, P_N)$ disagree, providing an indirect detector of symmetry-breaking. We model symmetry-breaking transitions as belonging to the upper tail of the empirical distribution of disagreement scores $d(s,a)$ and use these high-disagreement samples to define pseudo-labels $y(s,a) \in \{0,1\}$ for training $\lambda_\omega$ with a binary cross-entropy loss:

---

**Algorithm 1** Partially Equivariant Reinforcement Learning (PERL)

---

**Require:** Replay buffer $\mathcal{D}$, critics $Q_E, Q_N$, policies $\pi_E, \pi_N$
**Require:** Dynamics predictors $\hat{P}_E, \hat{P}_N$, gating functions $\lambda_\omega, \lambda_\zeta$, targets $\bar{Q}, \bar{\lambda}_\omega$
 1: Initialize all networks
 2: Initialize running statistics $(\mu, \sigma)$ for disagreement
 3: **for** $t = 1$ to $T$ **do**
 4:     Sample $a_t \sim \pi_\phi(\cdot \mid s_t)$                                    ▷ gated policy from Eq. (7)
 5:     Store $(s_t, a_t, r_t, s_{t+1})$ in $\mathcal{D}$
 6:     Train predictors $\hat{P}_E$ and $\hat{P}_N$ to minimize the predictive loss $\mathcal{L}_{\text{pred}}$        ▷ see App. B.1
 7:     Compute disagreement $d(s,a) = D\big(\hat{P}_E(\cdot \mid s,a), \hat{P}_N(\cdot \mid s,a)\big)$     ▷ see Sec. 6.1, App. B.1
 8:     Update the running statistics $(\mu, \sigma)$ over the disagreement $d(s,a)$
 9:     Update $\lambda_\omega$ and $\lambda_\zeta$ using BCE-loss (Eq. (5)) and expectile regression (Eq. (8))
10:     Update the critics with the objective (Eq. (9))
11:     Update the actor with the objective (Eq. (10))        ▷ SAC only; DQN uses greedy $\arg\max$
12:     Soft update $\bar{Q}$ and $\bar{\lambda}_\omega$
13: **end for**

---

$$\mathcal{L}_\lambda(\omega) = \mathbb{E}_{(s,a)\sim\mathcal{D}}\big[-y(s,a)\log\lambda_\omega(s,a) - (1-y(s,a))\log(1-\lambda_\omega(s,a))\big], \tag{5}$$

where $\mathcal{D}$ is the replay buffer. The gating network is optimized only through Eq. (5); during RL updates of the value function and policy, we treat $\lambda_\omega$ as fixed and do not backpropagate RL gradients into its parameters (see Appendix B.1 for implementation details).

## 6.2 PARTIALLY EQUIVARIANT REINFORCEMENT LEARNING

We couple the learned gating function to the critic and the actor, thereby implementing the PI-MDP framework under function approximation while training entirely in the true environment $\mathcal{M}_N$.

**Gated value mixtures under the true MDP.**    We parameterize the critic as a gated mixture:

$$Q_\theta(s,a) = \big(1 - \lambda_\omega(s,a)\big) Q_{E,\theta}(s,a) + \lambda_\omega(s,a) Q_{N,\theta}(s,a), \tag{6}$$

where $Q_E$ is an equivariant critic constrained by group symmetries and $Q_N$ is an unconstrained critic with no symmetry bias. The gating function $\lambda_\omega : \mathcal{S} \times \mathcal{A} \to \{0,1\}$ routes between the two networks. Conditioned on the binary gating $\lambda_\omega$ (cached per minibatch and used with stop-gradient), our TD-based critic (e.g., DQN, SAC) learns under $\mathcal{M}_N$ the best approximation within this mixed hypothesis class. With binary gating, the mixture reduces to a hard switch, activating either $Q_E$ or $Q_N$ depending on whether the state–action lies in a symmetric or symmetry-breaking region.

**Idealized compatibility (binary oracle gating).**    If $\lambda(s,a) \in \{0,1\}$ *perfectly* separates symmetric from broken regions and, on symmetric regions, the averaged dynamics coincide $(P_E, R_E) = (P_N, R_N)$, then the partially group-invariant operator $\mathcal{T}_H$ is identical to the true operator $\mathcal{T}_N$. In this idealized case, our TD targets exactly match $(\mathcal{T}_H Q)(s,a)$ and the mixture recovers the interpolating solution in Theorem 1. This motivates the use of $\lambda$ as a "local oracle" for symmetry-breaking. In practice, we approximate this oracle by the learned gating function $\lambda_\omega$, producing binary decisions as described above.

**Gated policy and actor gating function.**    For the policy, we employ a state-only gating function $\lambda_\zeta : \mathcal{S} \to \{0,1\}$ and define a product-of-experts (PoE) blend

$$\pi_\phi(\cdot \mid s) \propto \pi_{E,\phi}(\cdot \mid s)^{1-\lambda_\zeta(s)} \pi_{N,\phi}(\cdot \mid s)^{\lambda_\zeta(s)}. \tag{7}$$

This form naturally arises from SAC policy improvement: given the critic mixture $Q_\theta = (1 - \lambda_\omega)Q_E + \lambda_\omega Q_N$, the information projection in SAC yields a PoE between the energy models $\exp(Q_E/\alpha)$ and $\exp(Q_N/\alpha)$ (see Appendix A.4 for details). While a fully state–action gate in $\pi$ would be theoretically appealing, it is intractable in practice because the normalization constant of Eq. (7) would depend on $a$. Instead, we use a state-only gate $\lambda_\zeta(s)$, which is aligned with the critic gating function via a conservative aggregation loss. This conservativeness is crucial: since

symmetry-breaking may occur only for a subset of actions, $\lambda_\zeta(s)$ should activate whenever *any* action at state $s$ is flagged by $\lambda_\omega(s,a)$. This conservative choice does not compromise optimality, as taking the maximum ensures that any critical symmetry-breaking is accounted for while leaving the optimal policy unchanged.

$$\mathcal{L}_\lambda(\zeta) = \mathbb{E}_{(s,a)\sim\mathcal{D}}\Big[L_\tau\big(\lambda_\omega(s,a) - \lambda_\zeta(s)\big)\Big], \tag{8}$$

where $L_\tau$ is the expectile loss (Kostrikov et al., 2022). Taking $\tau \to 1$ approximates the $\max_a$ operator, ensuring that $\lambda_\zeta(s)$ conservatively reflects the maximum symmetry-breaking signal across actions. Per sample, Eq. (7) thus collapses to a hard switch between $\pi_E$ and $\pi_N$, retaining interpretability and computational tractability (details in Appendix B.2).

Besides the learned state-dependent gate $\lambda_\zeta(s)$, we also consider a sampled-max variant that approximates $\max_a \lambda_\omega(s,a)$ by taking the maximum over $\lambda_\omega(s,a_i)$ for $K$ sampled actions. In our ablations (Fig. 6), $K \in \{4,8\}$ performs similarly to $\lambda_\zeta(s)$, making this a reasonable choice for a lighter architecture, though the learned state gate is slightly more robust when symmetry-breaking is sparse and easy to miss with a few sampled actions.

**Training.** We train $Q_\theta$ and $\pi_\phi$ using standard objectives from deep RL: DQN (Mnih et al., 2013) for value-based methods and SAC (Haarnoja et al., 2018) for actor–critic methods, substituting in our gated parameterizations. In this way, the partially equivariant framework is realized within standard off-the-shelf algorithms, while the gates $\lambda_\omega$ and $\lambda_\zeta$ provide adaptive control over when equivariance is exploited and when it is suppressed.

$$J_Q(\theta) = \mathbb{E}_{(s,a,r,s')\sim\mathcal{D}} \tfrac{1}{2}\Big(Q_\theta(s,a) - r + \gamma\max_{a'} Q_{\bar\theta}(s',a')\Big)^2, \tag{9}$$

where $\bar\theta$ denotes target parameters and, $Q_\theta(s,a) = (1-\lambda_\omega(s,a))\,Q_{E,\theta}(s,a) + \lambda_\omega(s,a)\,Q_{N,\theta}(s,a)$.

$$J_\pi(\phi) = \mathbb{E}_{\substack{s\sim\mathcal{D}\\\epsilon\sim\mathcal{N}(0,I)}}\Big[\alpha\log\pi_\phi(a\mid s) - \min_{i=1,2} Q_{\theta_i}(s,a)\Big], \qquad a = \tanh\big(g_\phi(s,\epsilon)\big). \tag{10}$$

where $\alpha$ is the entropy temperature used in SAC (Haarnoja et al., 2018), and $\log\pi_\phi(a\mid s) = (1-\lambda_\zeta(s))\log\pi_{E,\phi}(a\mid s) + \lambda_\zeta(s)\log\pi_{N,\phi}(a\mid s)$. Please refer to Algorithm 1 for the pseudocode, and Appendix B for more details.

For the network architecture, we use separate trunks for the critics, the policy, and the one-step predictors, which provides stable training and a clean separation between equivariant and non-equivariant components. We also explored several trunk-sharing variants for parameter efficiency, but they did not consistently improve performance and sometimes harmed stability (see Fig. 8).

# 7 EXPERIMENTS

Our experiments aim to answer two main questions: (1) How does our method compare in terms of sample efficiency against the conventional RL and strictly equivariant methods? (2) How robust is our method to symmetry-breaking, relative to the other approximately equivariant approaches?

## 7.1 EXPERIMENTAL SETUP

We evaluate across discrete and continuous control settings: (1) **Grid-World** for intuitive analysis, and (2) **Locomotion** and **Manipulation** benchmarks using MuJoCo (Brockman et al., 2016), Fetch Reach (Plappert et al., 2018), and the UR5e manipulator (Tassa et al., 2018; Chuang, 2023). To systematically test robustness, we introduce fixed obstacles in the Grid-World that cause the transition kernel to deviate from ideal rotational symmetry. In the continuous tasks, symmetry-breaking naturally arises from realistic constraints like contacts and joint limits. We compare PE-DQN and PE-SAC against vanilla RL, strictly equivariant methods, and approximately equivariant baselines (Finzi et al., 2021a; Park et al., 2025). Details on the environment are provided in Appendix D.

**Grid-World.** We use a discrete $C_4$-symmetric gridworld as a lightweight testbed for analyzing robustness to symmetry-breaking. The MDP state consists only of the agent and goal coordinates;

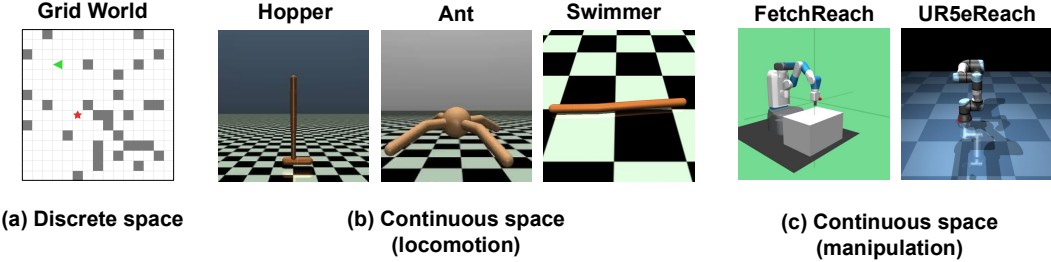

Figure 2: **Benchmark environments.** We evaluate our method across both discrete and continuous control tasks under symmetry-breaking conditions. Specifically, we use the Grid-World environment for the discrete case, and locomotion and manipulation tasks for the continuous case.

obstacles are part of the environment layout but are not encoded in this state representation. Symmetry is broken by placing fixed obstacles that cause the induced transition kernel $P_N(s' \mid s, a)$ over these coordinates to deviate from the ideal rotational symmetry. By varying the number of obstacles, we control the degree of symmetry-breaking and can clearly examine how PE-DQN adapts as the extent of symmetry violation increases.

To study reward-level symmetry-breaking, we also consider a variant in which a subset of obstacles becomes passable but returns a negative reward when traversed, leaving the transition structure unchanged while altering the reward function. Finally, we construct a stochastic variant with numerous obstacles, in which actions result in randomized neighboring transitions, creating complex dynamics that test the robustness of our disagreement-based gating under complex dynamics.

**Locomotion.**   We evaluate on continuous-control MuJoCo benchmarks using the same symmetry specifications as RPP (Finzi et al., 2021a), which include both exact and approximate symmetries. This setting allows us to test whether PE-SAC can extend the sample-efficiency benefits of equivariance from discrete Grid-World to challenging continuous-control tasks, while remaining robust to symmetry-breaking factors such as external forces or reward perturbations. All baselines are trained with SAC.

**Manipulation.**   We evaluate in manipulation settings, considering two reach tasks with $SO(3)$ symmetry. Fetch Reach serves as a simpler case, where the end-effector is constrained perpendicular to the floor and the goal is specified only by $(x, y, z)$ position. In contrast, UR5e Reach allows free end-effector orientation in addition to position, with a goal specified as an $SE(3)$ pose that includes both position and orientation. The inclusion of orientation control makes the task more representative of real-world manipulators. This progression from Fetch to UR5e enables us to test whether PE-SAC scales from constrained to more realistic manipulation scenarios. Symmetry-breaking naturally arises from collisions, floor contacts, and kinematic singularities. All methods use the same SAC backbone for comparability.

## 7.2  ANALYSIS

In Fig. 3, we show returns in Grid-World as the number of obstacles increases. When no symmetry-breaking factors are present, PE-DQN quickly converges to $\lambda \approx 0$ and behaves like a purely equivariant agent, matching the performance of strictly equivariant DQN. As obstacles are added, strictly equivariant DQN degrades much more rapidly than the other baselines, while approximately equivariant methods offer only minor gains over vanilla DQN in both sample efficiency and final return. In contrast, PE-DQN maintains strong performance across all obstacle counts, indicating robustness to localized symmetry-breaking and aligning with our theory that value errors remain controlled by $\epsilon_R$ and $\epsilon_P$ when the gate routes away from equivariance in mismatched regions.

Fig. 4 evaluates Grid-World variants that isolate reward-level symmetry-breaking and complex dynamics. For reward-level symmetry-breaking, we augment each predictor with a reward head $\hat{R}_i(s, a), \; i \in \{N, E\}$ and define disagreement as the sum of transition- and reward-level terms

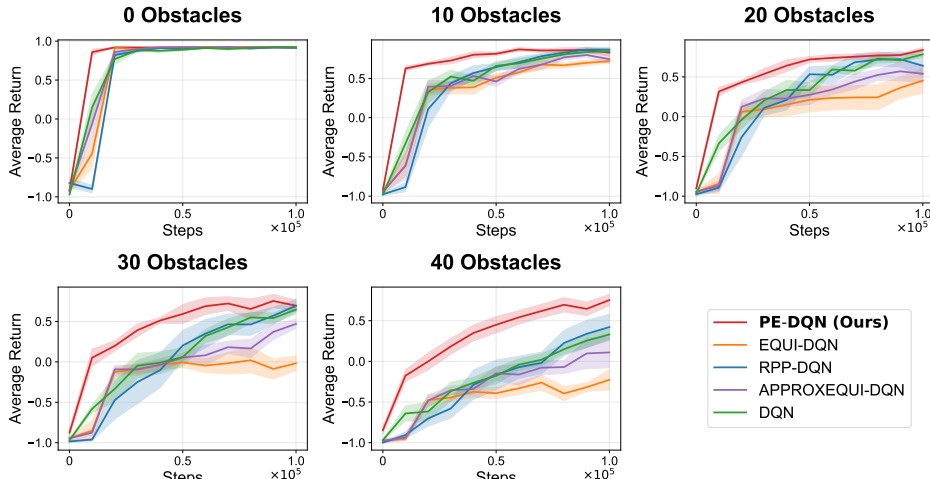

Figure 3: **Performance comparison in the discrete space (Grid-World) environment.** We evaluate the average performance over 100K steps with five random seeds. Shaded regions denote standard error. We vary the number of obstacles, which act as symmetry-breaking factors. PE-DQN consistently outperforms the baselines, and the performance gap widens as symmetry-breaking increases, demonstrating both robustness and sample efficiency.

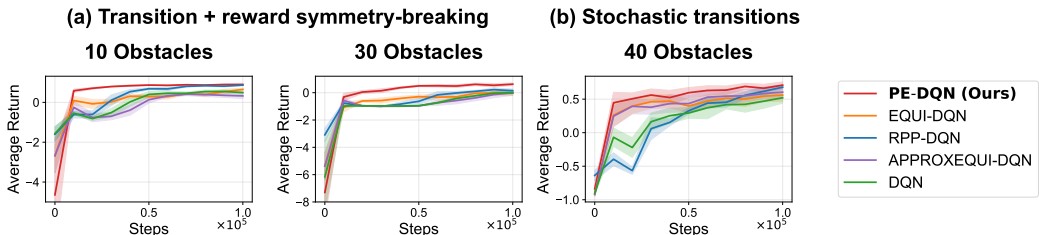

Figure 4: **Performance comparison in Grid-World under reward-level symmetry-breaking and complex dynamics.** Results are averaged over 100K environment steps with five random seeds; shaded regions denote standard error. **(a)** Reward-level symmetry-breaking is introduced by making half of the obstacles passable while assigning a negative reward upon traversal, in layouts with 10 and 30 obstacles. **(b)** Complex dynamics setting with stochastic transitions in 40-obstacle layout. PE-DQN consistently outperforms the baselines in both settings, indicating robustness to reward-level symmetry-breaking and challenging dynamics.

(Appendix B.1). This setting converts a subset of obstacles into passable but penalized cells, so symmetry is broken purely through the reward while the transition structure remains unchanged. Across the 10- and 30-obstacle layouts, PE-DQN consistently achieves the highest returns, with RPP-DQN as the strongest baseline but still trailing in both sample efficiency and final performance.

In the complex-dynamics variant with 40 obstacles and stochastic transitions, random slips partially mask transition-level symmetry-breaking (e.g., the agent can occasionally bypass obstacles by chance), which allows strictly equivariant and Approximately Equivariant DQN to recover reasonable performance. Nonetheless, PE-DQN continues to attain the best returns and learning speed, indicating that the disagreement-based gate remains effective even when the underlying dynamics are noisy and harder to model.

In Fig. 5, we report results on continuous-control locomotion and manipulation tasks. In Hopper, PE-SAC learns faster than all baselines and reaches a strong plateau, although vanilla SAC attains a slightly higher asymptotic return. In Ant, PE-SAC clearly dominates in both sample efficiency and final performance. In Swimmer, where symmetry is nearly exact, the strictly equivariant and Ap-

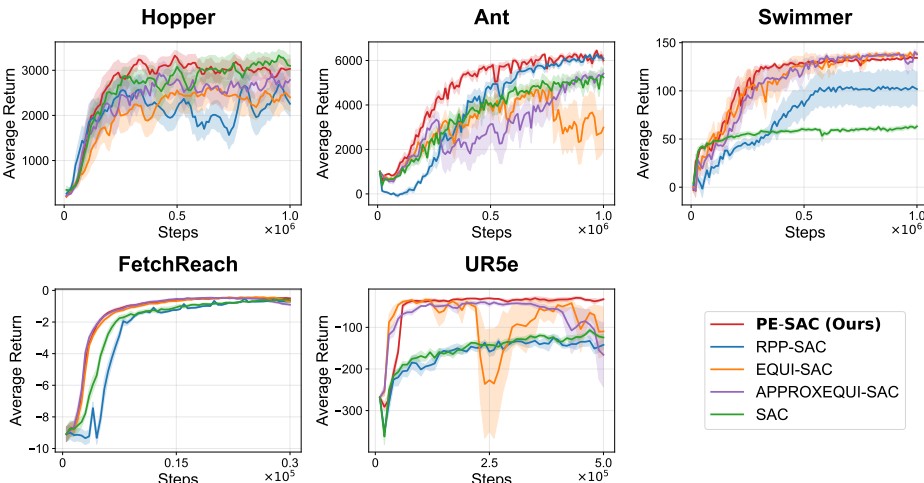

Figure 5: **Performance comparison in the continuous space environments.** Results are averaged over 1M training steps in MuJoCo tasks, and 30K, 500K steps in the Fetch, UR5e Reach environment, using eight random seeds from locomotion tasks and five random seeds from manipulation tasks. Shaded regions denote standard error. For RPP (Finzi et al., 2021a), we re-ran the official code. Discrepancies with the reported numbers arise because RPP reports "max over steps" rather than average performance. PE-SAC consistently outperforms all baselines across these tasks.

proximately Equivariant SAC baselines achieve the highest final returns, while PE-SAC converges quickly to a slightly lower but competitive level, reflecting the advantage of enforcing exact symmetry in this regime. In Fetch Reach, PE-SAC, exact equivariant SAC, and Approximately Equivariant SAC perform similarly. In UR5e Reach, where symmetry-breaking is substantial due to realistic dynamics and free orientation, the strictly equivariant and Approximately Equivariant SAC variants become unstable or collapse, whereas PE-SAC remains stable and attains the best overall returns by shifting toward the non-equivariant head.

Overall, these results support our central claim: by selectively mitigating local equivariance errors, PE-DQN and PE-SAC retain the sample efficiency benefits of equivariance in symmetric regions while remaining robust in symmetry-broken regimes across discrete, continuous, and realistic robotic environments.

# 8 CONCLUSION

In this work, we introduced the **PI-MDP**, a framework that mitigates global error propagation from local symmetry-breaking. Building on this foundation, we developed **Partially Equivariant RL (PE-RL)** algorithms—PE-DQN for discrete control and PE-SAC for continuous control—that consistently improved sample efficiency and robustness over conventional RL, exact-equivariant methods, and approximate baselines.

The main limitation is computation: auxiliary predictors and gates increase training time. Furthermore, under pervasive symmetry-breaking (e.g., gravity), the framework largely defaults to non-equivariant networks, offering marginal benefits over standard RL. However, since symmetry-breaking in typical control tasks is localized, our approach successfully preserves both sample efficiency and robustness to symmetry-breaking.

Future work includes extending PE-RL to vision-based control, advancing the practicality of symmetry-aware reinforcement learning for real-world continuous control.

ACKNOWLEDGMENTS

We would like to thank Hyunwoo Ryu for insightful discussions that helped improve this manuscript. This work was supported by the National Research Foundation of Korea (NRF) grant funded by the Korea government (MSIT) (No.RS-2024-00344732). This work was also supported by the Korea Institute of Science and Technology (KIST) Institutional Program (Project No.2E33801-25-015), the Institute of Information & communications Technology Planning & Evaluation (IITP) grant funded by the Korea government (MSIT) (No. RS-2024-00457882, AI Research Hub Project), and the Institute of Information & Communications Technology Planning & Evaluation (IITP) grant (RS-2020-II201361, Artificial Intelligence Graduate School Program (Yonsei University)). Joohwan Seo and Roberto Horowitz are funded by the Hong Kong Center for Construction Robotics Limited (HKCRC).

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

# A  THEORETICAL PROOFS

## A.1  PROOF OF LEMMA 1 AND PROPOSITION 1

**Lemma 1** (One-step Bellman error). *For any bounded $Q$ and any $(s,a) \in \mathcal{S} \times \mathcal{A}$,*

$$\big|(\mathcal{T}_N Q)(s,a) - (\mathcal{T}_E Q)(s,a)\big| \leq \epsilon_R(s,a) + 2\gamma \|V_Q\|_\infty \epsilon_P(s,a).$$

*Proof.* By the triangle inequality,

$$\big|(\mathcal{T}_N Q)(s,a) - (\mathcal{T}_E Q)(s,a)\big|$$
$$= \Big| R_N(s,a) - R_E(s,a) + \gamma\big(\mathbb{E}_{s' \sim P_N(\cdot|s,a)}[V_Q(s')] - \mathbb{E}_{s' \sim P_E(\cdot|s,a)}[V_Q(s')]\big)\Big|$$
$$\leq \epsilon_R(s,a) + \gamma\Big|\mathbb{E}_{P_N}[V_Q] - \mathbb{E}_{P_E}[V_Q]\Big|.$$

Using the total-variation inequality $\big|\mathbb{E}_P[f] - \mathbb{E}_Q[f]\big| \leq 2\|f\|_\infty \, \mathrm{TV}(P,Q)$ with the definition of $\epsilon_P$ in Eq. (1) and $f = V_Q$,

$$\big|\mathbb{E}_{P_N}[V_Q] - \mathbb{E}_{P_E}[V_Q]\big| \leq 2\|V_Q\|_\infty \epsilon_P(s,a).$$

Combining the bounds leads to the lemma. $\qquad\square$

**Proposition 1.** *(Value-function gap). Let $Q_i^*$ be the optimal action–value function in MDP $i$. Then,*

$$\|Q_N^* - Q_E^*\|_\infty \leq \frac{1}{1-\gamma}\|\delta\|_\infty.$$

*Proof.* Since $Q_N^* = \mathcal{T}_N Q_N^*$ and $Q_E^* = \mathcal{T}_E Q_E^*$, we have

$$\|Q_N^* - Q_E^*\|_\infty = \|\mathcal{T}_N Q_N^* - \mathcal{T}_E Q_E^*\|_\infty \leq \|\mathcal{T}_N Q_N^* - \mathcal{T}_N Q_E^*\|_\infty + \|\mathcal{T}_N Q_E^* - \mathcal{T}_E Q_E^*\|_\infty.$$

The Bellman optimality operator is a $\gamma$-contraction in the sup norm, so

$$\|\mathcal{T}_N Q_N^* - \mathcal{T}_N Q_E^*\|_\infty \leq \gamma\|Q_N^* - Q_E^*\|_\infty.$$

By Lemma 1 applied with $Q = Q_E^*$ and the bounded $\|V_{Q_E^*}\|_\infty \leq V_{\max}$, we have

$$\|\mathcal{T}_N Q_E^* - \mathcal{T}_E Q_E^*\|_\infty \leq \|\delta\|_\infty,$$

where $\delta(s,a)$ represents the pointwise one-step approximation error defined in Eq. (2). Combining the two inequalities gives

$$\|Q_N^* - Q_E^*\|_\infty \leq \gamma\|Q_N^* - Q_E^*\|_\infty + \|\delta\|_\infty.$$

Rearranging results in

$$\|Q_N^* - Q_E^*\|_\infty \leq \frac{1}{1-\gamma}\|\delta\|_\infty,$$

which completes the proof. $\qquad\square$

## A.2  PROOF OF THEOREM 1 AND COROLLARY 1

**Theorem 1** (Partially group-invariant optimality operator). *Let $\mathcal{T}_i$ denote the (hard) Bellman optimality operator in MDP $i \in \{E, N, H\}$, $(\mathcal{T}_i Q)(s,a) = R_i(s,a) + \gamma\, \mathbb{E}_{s' \sim P_i(\cdot|s,a)}[\max_{a'} Q(s',a')]$. For any bounded $Q : \mathcal{S} \times \mathcal{A} \to \mathbb{R}$ and all $(s,a)$,*

$$(\mathcal{T}_H Q)(s,a) = (1 - \lambda(s,a))(\mathcal{T}_E Q)(s,a) + \lambda(s,a)(\mathcal{T}_N Q)(s,a). \tag{11}$$

*If $|R_E|, |R_N| \leq R_{\max}$ and $\gamma \in (0,1)$, then $\mathcal{T}_H$ is a $\gamma$-contraction and admits a unique fixed point $Q_H^*$.*

*Proof. Identity Eq. (11).* By Definition 2, for any $(s, a)$,

$$(\mathcal{T}_H Q)(s, a) = (1 - \lambda(s, a))\Big(R_E(s, a) + \gamma \mathbb{E}_{s' \sim P_E(\cdot | s, a)}\big[\max_{a'} Q(s', a')\big]\Big)$$
$$+ \lambda(s, a)\Big(R_N(s, a) + \gamma \mathbb{E}_{s' \sim P_N(\cdot | s, a)}\big[\max_{a'} Q(s', a')\big]\Big),$$

which equals $(1 - \lambda)\mathcal{T}_E Q + \lambda \mathcal{T}_N Q$ pointwise.

*Contraction.* Let $Q_1, Q_2$ be bounded. Using Eq. (3) and that $\mathcal{T}_E, \mathcal{T}_N$ are $\gamma$-contractions,

$$\big|\mathcal{T}_H Q_1(s, a) - \mathcal{T}_H Q_2(s, a)\big|$$
$$= \Big|(1 - \lambda(s, a))\big(\mathcal{T}_E Q_1(s, a) - \mathcal{T}_E Q_2(s, a)\big) + \lambda(s, a)\big(\mathcal{T}_N Q_1(s, a) - \mathcal{T}_N Q_2(s, a)\big)\Big|$$
$$\leq (1 - \lambda(s, a))\,\|\mathcal{T}_E Q_1 - \mathcal{T}_E Q_2\|_\infty + \lambda(s, a)\,\|\mathcal{T}_N Q_1 - \mathcal{T}_N Q_2\|_\infty$$
$$\leq \gamma\,\|Q_1 - Q_2\|_\infty.$$

Taking the supremum over $(s, a)$ gives

$$\|\mathcal{T}_H Q_1 - \mathcal{T}_H Q_2\|_\infty \leq \gamma\,\|Q_1 - Q_2\|_\infty.$$

Bounded rewards ensure $\mathcal{T}_H$ maps bounded $Q$ into bounded $Q$. By Banach's fixed point theorem, $\mathcal{T}_H$ has a unique fixed point $Q_H^*$. $\qquad\square$

**Corollary 1** (Proximity bound). *Let $Q_N^*$ be the optimal action–value of the true MDP $\mathcal{M}_N$, and let $\delta(s, a)$ be the one-step pointwise Bellman error bound. Then*

$$\|Q_H^* - Q_N^*\|_\infty \;\leq\; \frac{1}{1 - \gamma}\,\Big\|(1 - \lambda)\,\delta\Big\|_\infty. \tag{12}$$

*Proof.*

$$\|Q_H^* - Q_N^*\|_\infty = \|\mathcal{T}_H Q_H^* - \mathcal{T}_N Q_N^*\|_\infty$$
$$\leq \|\mathcal{T}_H Q_H^* - \mathcal{T}_H Q_N^*\|_\infty + \|\mathcal{T}_H Q_N^* - \mathcal{T}_N Q_N^*\|_\infty$$
$$\leq \gamma\|Q_H^* - Q_N^*\|_\infty + \|(1 - \lambda)\,(\mathcal{T}_E Q_N^* - \mathcal{T}_N Q_N^*)\|_\infty.$$

Expanding pointwise,

$$(\mathcal{T}_E Q_N^* - \mathcal{T}_N Q_N^*)(s, a) = (R_E - R_N)(s, a) + \gamma\Big(\mathbb{E}_{s' \sim P_E(\cdot | s, a)}[V_N(s')] - \mathbb{E}_{s' \sim P_N(\cdot | s, a)}[V_N(s')]\Big).$$

By the definition of total variation distance,

$$\big|\mathbb{E}_{P_E}[V_N] - \mathbb{E}_{P_N}[V_N]\big| \;\leq\; 2\,\epsilon_P(s, a)\,V_{\max},$$

where $V_{\max} = R_{\max}/(1 - \gamma)$, $\epsilon_P(s, a) = \frac{1}{2}\int_{\mathcal{S}} |P_N(s' | s, a) - P_E(s' | s, a)|\,ds'$ introduced in Eq. (1), and $\delta(s, a)$ defined in Eq. (2). Rearranging gives Eq. (12) $\qquad\square$

### A.3 PARTIAL GROUP-INVARIANCE IN SOFT MDPS

Since the PI-MDP defined in Definition 2 is a valid MDP, the soft policy iteration framework (Haarnoja et al., 2018) applies unchanged. We show the evaluation identity and the standard improvement step for completeness.

**Policy evaluation.** For a fixed policy $\pi$, define the soft state value $V_Q^\pi(S) := \mathbb{E}_{a \sim \pi(\cdot | s)}\big[Q(s, a) - \alpha \log \pi(a \mid s)\big]$ with temperature $\alpha > 0$. The soft Bellman operator under $\mathcal{M}_H$ is

$$(\mathcal{T}_H^\pi Q)(s, a) = R_H(s, a) + \gamma \mathbb{E}_{s' \sim P_H(\cdot | s, a)}\big[V_Q^\pi(s')\big].$$

Writing $\lambda := \lambda(s, a)$ for brevity, $R_H$ and $P_H$ (Definition 2) leads to the pointwise identity

$$(\mathcal{T}_H^\pi Q)(s, a) = (1 - \lambda)\,R_E(s, a) + \lambda\,R_N(s, a)$$
$$+ \gamma\Big((1 - \lambda)\,\mathbb{E}_{s' \sim P_E(\cdot | s, a)}\big[V_Q^\pi(s')\big] + \lambda\,\mathbb{E}_{s' \sim P_N(\cdot | s, a)}\big[V_Q^\pi(s')\big]\Big)$$
$$= (1 - \lambda)\,(\mathcal{T}_E^\pi Q)(s, a) + \lambda\,(\mathcal{T}_N^\pi Q)(s, a).$$

Thus, soft evaluation under $\mathcal{M}_H$ is the same convex combination of the component evaluation as in hard (max) case.

**Policy improvement.** Treating $\lambda$ as fixed, the soft policy improvement step follows the SAC formulation:

$$\pi_{k+1}(\cdot \mid s) = \arg\min_{\pi} \ D_{\mathrm{KL}}\left(\pi(\cdot \mid s) \,\Big\|\, \frac{\exp\big(Q^{\pi_k}(s,\cdot)/\alpha\big)}{Z_k(s)}\right), \tag{13}$$

where $Z_k(s)$ is the normalizing constant. Alternating evaluation under $(\mathcal{T}_H^{\pi})$ and the update Eq. (13) is exactly soft policy iteration on $\mathcal{M}_H$. Under the standard assumptions of Haarnoja et al. (2018), this admits a unique soft fixed point and corresponding policy.

### A.4 POLICY PARAMETERIZATION AND TRACTABILITY FOR PE-SAC

**PoE from SAC policy improvement.** For a fixed gating function $\lambda : \mathcal{S} \times \mathcal{A} \to [0,1]$ and $Q_\theta = (1-\lambda)Q_E + \lambda Q_N$, the SAC information projection (for each $s$)

$$\pi^*(\cdot \mid s) = \arg\min_{\pi} D_{\mathrm{KL}}\left(\pi(\cdot \mid s) \,\Big\|\, \frac{\exp\big(Q_\theta(s,\cdot)/\alpha\big)}{Z_\theta(s)}\right)$$

has a unique solution

$$\pi^*(a \mid s) \ \propto \ \exp\left(\frac{(1-\lambda)Q_E(s,a)+\lambda Q_N(s,a)}{\alpha}\right)$$
$$= \big[\exp\big(Q_E(s,a)/\alpha\big)\big]^{1-\lambda(s,a)}\big[\exp\big(Q_N(s,a)/\alpha\big)\big]^{\lambda(s,a)}.$$

If $\lambda$ is state-only, $\lambda = \lambda(s)$, then the normalizers of $\exp(Q_E/\alpha)$ and $\exp(Q_N/\alpha)$ are constant in $a$ and factor out, leading to the geometric mixture of normalized policies:

$$\pi^*(\cdot \mid s) \ \propto \ \pi_E(\cdot \mid s)^{1-\lambda(s)} \pi_N(\cdot \mid s)^{\lambda(s)}$$

where

$$\pi_E(\cdot \mid s) \propto \exp\big(Q_E(s,\cdot)/\alpha\big), \ \ \pi_N(\cdot \mid s) \propto \exp\big(Q_N(s,\cdot)/\alpha\big).$$

**Why an action-dependent gating function breaks reparameterization.** Write the energies $f_E := Q_E/\alpha$ and $f_N := Q_N/\alpha$. Define the unnormalized density

$$u_\phi(a \mid s) \ := \ \exp\big\{(1-\lambda(s,a))\, f_E(s,a) + \lambda(s,a)\, f_N(s,a)\big\}, \qquad Z_\phi(s) := \int_{\mathcal{A}} u_\phi(a \mid s)\, da.$$

When $\lambda = \lambda(s,a)$, the normalizer $Z_\phi(s)$ has no closed form and its gradient with respect to the parameters inside $\lambda, f_E, f_N$ is intractable. Therefore,

$$\log \pi_\phi(a \mid s) \ = \ (1-\lambda)f_E(s,a) + \lambda f_N(s,a) \ - \ \log Z_\phi(s)$$

cannot be evaluated with a tractable pathwise sampler $a = g_\phi(s,\epsilon)$, so the reparameterized SAC actor objective

$$J(\phi) \ = \ \mathbb{E}_{s,\epsilon}\big[\alpha \log \pi_\phi(a \mid s) - Q_\theta(s,a)\big]$$

is not tractable. This motivates a *state-only* gating function in the actor.

**Gaussian policy with squashing (state-only gating).** Following SAC, we use an unbounded Gaussian for a pre-squash variable $u \in \mathbb{R}^D$ and apply an elementwise $\tanh$ to obtain bounded actions $a = \tanh(u)$. Let the two pre-squash Gaussian *densities* be

$$p_E(u \mid s) = \mathcal{N}\big(u;\, \mu_E(s), \Sigma_E(s)\big), \qquad p_N(u \mid s) = \mathcal{N}\big(u;\, \mu_N(s), \Sigma_N(s)\big),$$

and let the gating function be state-only, $\lambda = \lambda(s) \in [0,1]$. Define the unnormalized product

$$\tilde{p}_H(u \mid s) \ := \ p_E(u \mid s)^{1-\lambda(s)}\, p_N(u \mid s)^{\lambda(s)}.$$

Since the exponents are constants for fixed $s$, $\tilde{p}_H$ is proportional to a Gaussian. In particular,

$$p_H(u \mid s) = \mathcal{N}\big(u;\, \mu_H(s), \Sigma_H(s)\big),$$
$$\Sigma_H^{-1}(s) = (1-\lambda(s))\, \Sigma_E^{-1}(s) + \lambda(s)\, \Sigma_N^{-1}(s), \tag{14}$$
$$\mu_H(s) = \Sigma_H(s)\Big((1-\lambda(s))\, \Sigma_E^{-1}(s)\mu_E(s) + \lambda(s)\, \Sigma_N^{-1}(s)\mu_N(s)\Big).$$

With $a = \tanh(u)$ and the change-of-variables formula (cf. SAC(Haarnoja et al., 2018), Eqs. (20)–(21)),

$$\pi_H(a \mid s) = p_H(u \mid s) \left| \det\left(\frac{\partial a}{\partial u}\right) \right|^{-1}$$

$$\log \pi_H(a \mid s) = \log p_H(u \mid s) - \sum_{i=1}^{D} \log\left(1 - \tanh^2(u_i)\right),$$

where $u = \operatorname{arctanh}(a)$ and the Jacobian $\partial a / \partial u$ is diagonal with entries $1 - \tanh^2(u_i)$. When the gating is binary, $\lambda(s) \in \{0, 1\}$, Eq. (14) reduces to the corresponding expert.

## B  IMPLEMENTATION DETAILS

### B.1  DETAILS FOR LEARNING $\lambda_\omega$ AND PREDICTORS $\hat{P}_E, \hat{P}_N$

**One-step predictors and disagreement metric.**  We train two one-step predictors on replay. In the environment with discrete state spaces (e.g., the grid-world with obstacles), the predictors parameterize the transition kernel $\hat{P}_i(s' \mid s, a)$ for $i \in \{E, N\}$, implemented as a categorical distribution over the possible next states. In environments with continuous state spaces, the predictors output the increment to the next state, $\hat{P}_i : (s, a) \mapsto \Delta\hat{s}_i(s, a)$, intended to approximate the transition dynamics of $\mathcal{M}_E$ and $\mathcal{M}_N$, respectively.

In the discrete case, each predictor is trained via a cross-entropy loss on transitions $(s, a, s')$.

$$\mathcal{L}^{(i)}_{\text{pred}} = \mathbb{E}_{(s,a,s')\sim\mathcal{D}}\left[-\log \hat{P}_i(s' \mid s, a)\right], \qquad i \in \{E, N\}.$$

The discrepancy measure $D(\hat{P}_E, \hat{P}_N)$ is defined as the total-variation distance, consistent with the definition of $\epsilon_P(s, a)$ in Eq. (1):

$$d(s, a) = \frac{1}{2}\sum_{s'\in\mathcal{S}}\left|\hat{P}_N(s' \mid s, a) - \hat{P}_E(s' \mid s, a)\right|.$$

In the continuous case, each predictor is optimized by minimizing mean squared error on the state increment $\Delta s := s' - s$:

$$\mathcal{L}^{(i)}_{\text{pred}} = \mathbb{E}_{(s,a,s')\sim\mathcal{D}}\left[\left\|\Delta\hat{s}_i(s, a) - \Delta s\right\|_2^2\right], \qquad i \in \{E, N\}.$$

The disagreement is then defined as the squared difference between predicted increments:

$$d(s, a) = \left\|\Delta\hat{s}_E(s, a) - \Delta\hat{s}_N(s, a)\right\|_2^2.$$

**One-step reward prediction and disagreement metric.**  In the variants of Grid-World experiments (Fig. 4), we additionally equip each predictor with a reward head $\hat{R}_i(s, a)$, $i \in \{E, N\}$, implemented as an extra head on top of the shared predictor trunk $\hat{P}_i$. In this case, the transition and reward predictors are trained jointly on $(s, a, r, s')$ with the combined loss

$$\mathcal{L}^{(i)}_{\text{pred}} = \mathbb{E}_{(s,a,r,s')\sim\mathcal{D}}\left[-\log \hat{P}_i(s' \mid s, a) + \left\|\hat{R}_i(s, a) - r\right\|_2^2\right], \qquad i \in \{E, N\}.$$

The reward component of the disagreement is defined as an $\ell_1$ distance, consistent with $\epsilon_R(s, a)$ in Eq. (1), and the overall disagreement combines transition and reward terms:

$$d(s, a) = \frac{1}{2}\sum_{s'\in\mathcal{S}}\left|\hat{P}_N(s' \mid s, a) - \hat{P}_E(s' \mid s, a)\right| + \left|\hat{R}_N(s, a) - \hat{R}_E(s, a)\right|.$$

**Disagreement thresholding and label generation.**  We maintain running statistics $(\mu_t, \sigma_t)$ of $d(s, a)$ via the Welford algorithm (Chan et al., 1983), form a raw threshold $\hat{\tau}_t = \mu_t + \kappa\sigma_t$ (reflecting the assumption that symmetry-breaking is sporadic), and then apply exponential smoothing:

$$\tau_t \leftarrow \beta\tau_{t-1} + (1 - \beta)\hat{\tau}_t.$$

Binary supervision is given by $y(s, a) = \mathbb{1}\{d(s, a) > \tau_t\}$. Here, $\kappa$ is a hyperparameter that controls what fraction of the disagreement distribution is treated as symmetry-breaking. We find that exhaustive tuning is unnecessary: a rough estimate of whether symmetry violations are sparse or dense is sufficient to choose a robust $\kappa$ (Fig. 9).

For more challenging dynamics (e.g., Grid-World environment with $20, 30,$ or $40$ obstacles), we found a slight performance gain from a batchwise quantile-based variant: within each minibatch $\mathcal{B}$, we treat $\{d(s, a) : (s, a) \in \mathcal{B}\}$ as an empirical distribution and set a batchwise threshold

$$\tau_{\mathcal{B}} = Q_\alpha\big(\{d(s, a) : (s, a) \in \mathcal{B}\}\big),$$

where $Q_\alpha$ denotes the upper $\alpha$-quantile. Binary labels are then $y(s, a) = \mathbb{1}\{d(s, a) > \tau_{\mathcal{B}}\}$.

**Gating function training and stochastic gating.** We train the gating network $\lambda_\omega : \mathcal{S} \times \mathcal{A} \to [0, 1]$ to estimate the likelihood of symmetry-breaking using the binary cross-entropy loss (Eq. (5)) on minibatches from the replay buffer $\mathcal{D}$. During each RL update of the value function and policy, we recompute and cache $\lambda_\omega(s, a)$ on the sampled minibatch and obtain a *stochastic hard gate* by Bernoulli sampling:

$$p(s, a) := \lambda_\omega(s, a), \qquad \tilde{\lambda}(s, a) \sim \text{Bernoulli}\big(p(s, a)\big).$$

We then form $Q_\theta = (1 - \tilde{\lambda})Q_E + \tilde{\lambda}Q_N$ for the critic update and use $\tilde{\lambda}$ for the actor-side alignment (Sec. 6.2). Gradients from $Q/\pi$ do *not* flow into $\omega$ (stop-gradient through $\tilde{\lambda}$).

**Target gate to reduce variance.** To reduce non-stationarity and variance induced by stochastic gating, we maintain an exponential moving average (EMA) of the gating probabilities for use in the target $Q$-updates:

$$\bar{p} \leftarrow \tau_\lambda \bar{p} + (1 - \tau_\lambda) p,$$

where $p = \lambda_\omega(s, a)$ denotes the current gate probability. The hard RL gate is then sampled from the smoothed probability $\bar{p}$ rather than from $p$:

$$\tilde{\lambda}(s, a) \sim \text{Bernoulli}\big(\bar{p}\big).$$

**Warm-start.** To avoid noisy labels before the dynamics predictors stabilize, we use a warm-up period $W$ steps during which the gate loss is disabled (i.e., $y \equiv 0$ and $\omega$ is not updated). A small prior routing is used by clamping $\tilde{\lambda}=1$ with probability $p_{\text{warm}}$ during warm-up.

### B.2 Details for learning $\lambda_\zeta(s)$

We train a state-only actor gate $\lambda_\zeta : \mathcal{S} \to [0, 1]$ to conservatively aggregate the action-dependent critic gate via

$$\lambda_\zeta(s) \approx \max_a \lambda_\omega(s, a).$$

To do so we adopt *expectile regression* with a high expectile level $\tau \to 1$, which approximates the max while remaining stable on in-distribution actions. Concretely, for each state $s$ we draw $M$ candidate actions $\{a_i\}_{i=1}^M$ (from current policies; see sampling details below) and minimize

$$\mathcal{L}_\lambda(\zeta) = \mathbb{E}_{s \sim \mathcal{D}}\left[ \frac{1}{M} \sum_{i=1}^M L_\tau\big(\lambda_\omega(s, a_i) - \lambda_\zeta(s; \zeta)\big) \right], \qquad L_\tau(u) = |\tau - \mathbb{1}\{u < 0\}|\, u^2.$$

This objective encourages $\lambda_\zeta(s)$ to match the upper tail of $\{\lambda_\omega(s, a_i)\}_{i=1}^M$, leading to a conservative state-level gate.

**Bernoulli actor gating (training & inference).** At both training and inference, we use a binary actor gating sampled from the probability $\lambda_\zeta(s)$:

$$\tilde{\lambda}_\zeta(s) \sim \text{Bernoulli}\big(\lambda_\zeta(s)\big).$$

For RL updates we cache $\tilde{\lambda}_\zeta$ per minibatch and apply stop-gradient through the sample.

**Candidate action sampling for expectiles.**   We form the candidate set $\{a_i\}_{i=1}^M$ per state by drawing from a mixture of current policies (e.g., $\pi_E, \pi_N$). This increases the chance of including symmetry-breaking actions. We use the same $M$ across tasks (see the hyperparameters Table 8).

**Warm-start.**   To avoid noisy supervision before $\lambda_\omega$ stabilizes, we apply a short warm-up period $W$ steps where $\mathcal{L}_\lambda(\zeta)$ is disabled; We use a small prior bias by clamping $\tilde{\lambda}_\zeta = 1$ with probability $p_{\text{warm}}$ during warm-up.

**Gradient isolation.**   Gradients from the RL losses do *not* flow into $\lambda_\zeta$; the gate is updated only via the expectile objective above.

### B.3   NETWORKS

For the Grid-World experiments, we implemented all equivariant networks from MDP-Homomorphic Networks (Van der Pol et al., 2020). In continuous control tasks, we used EMLP layers (Finzi et al., 2021b). The remaining networks, including $\pi_N$, $\hat{P}_N$, $\lambda_\omega$, $\lambda_\zeta$, and the critics, were implemented as standard MLPs.

### B.4   IMPLEMENTATION FRAMEWORK

Our implementation builds on the Residual Pathway Prior (RPP) codebase (Finzi et al., 2021a), which provides flexible infrastructure for combining equivariant and non-equivariant components. We extend this framework with our gated $Q$-networks, gated policies, and disagreement-based $\lambda$ supervision, while keeping the training loops and optimization settings consistent with RPP.

## C   ABLATION STUDIES

We present additional experiments on alternative actor-gating schemes, hard vs. soft gating, and shared trunks for $Q$, policy, and predictor networks for parameter efficiency. All curves are averaged over five random seeds, and shaded regions indicate standard error.

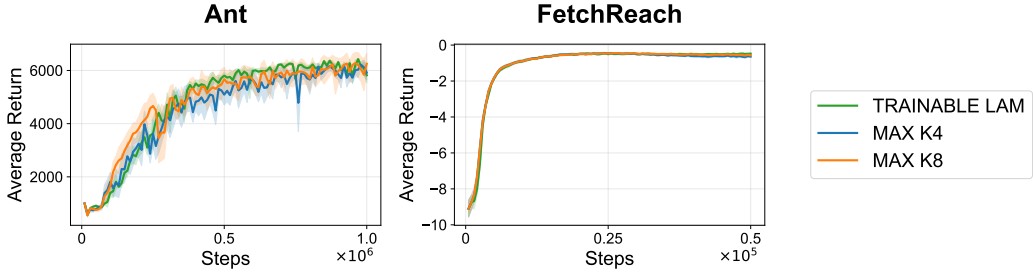

Figure 6: **Ablation on actor-gating schemes.** As described in Sec. 6.2, our main method uses a trainable state gate $\lambda_\zeta(s)$ to approximate the conservative aggregation $\max_a \lambda_\omega(s, a)$, so that symmetry-breaking can be detected even when it occurs only for a small subset of actions. Here we compare this gate to a simpler sampled-max variant that estimates $\max_a \lambda_\omega(s, a)$ by taking the maximum over $K$ actions, obtained by sampling $K/2$ actions from each of $\pi_E$ and $\pi_N$. In the legend, **TRAINABLE LAM** denotes the $\lambda_\zeta(s)$, and **MAX K4** and **MAX K8** denote these sampled-max schemes with $K \in \{4, 8\}$. All three options achieve similar performance and sample efficiency, indicating that the sampled-max gate is a reasonable alternative when architectural simplicity is prioritized.

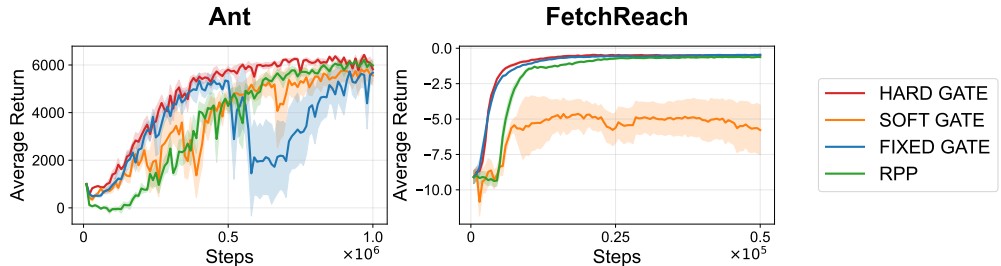

Figure 7: **Hard vs. soft gating in practice.** The theory permits any measurable gate $\lambda(s, a) \in [0, 1]$, as long as symmetry-breaking regions are routed to the true MDP so that the bound in Corollary 1 remains tight. In our implementation, however, $\lambda_\omega$ is trained separately from the critics, so using it as a soft mixing weight inside the Bellman backup can affect stability. We compare three variants: **HARD GATE**, our default, which samples a binary gate and routes entirely to $Q_E$ or $Q_N$; **SOFT GATE**, which uses $\lambda_\omega(s, a) \in [0, 1]$ directly as a convex-combination weight; **FIXED GATE**, which uses a constant mixture $\lambda(s, a) = 0.5$ (analogous to a fixed blend of equivariant and non-equivariant $Q$-heads); and **RPP** (Finzi et al., 2021a), a baseline that combines equivariant and non-equivariant features within the linear layers. On Ant, FIXED GATE exhibits noisy learning but eventually reaches reasonably good performance, while SOFT GATE is more stable but remains below HARD GATE (and RPP). On Fetch Reach, FIXED GATE performs comparably to HARD GATE and outperforms RPP, whereas SOFT GATE fails to learn, supporting our choice of hard gating in the practical algorithm.

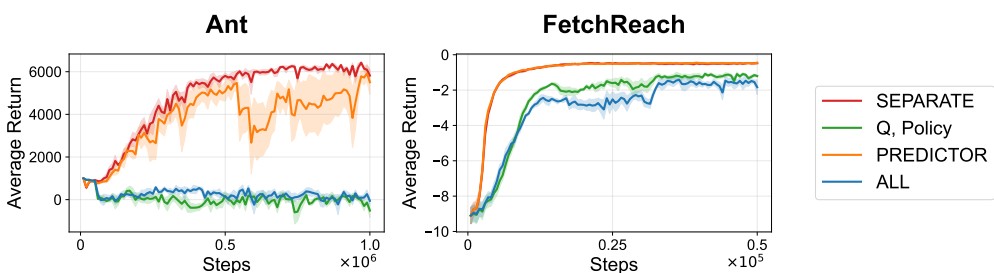

Figure 8: **Shared vs. separate trunks for $Q$, policy, and predictors.** We compare four architectures. **SEPARATE** is our default, with distinct networks for the critics, policy, and one-step predictors. **Q, POLICY** shares an equivariant trunk between the critics and the policy, followed by equivariant and non-equivariant heads for $(Q_E, Q_N)$ and $(\pi_E, \pi_N)$. **PREDICTOR** shares an equivariant trunk between the predictors $\hat{P}_E$ and $\hat{P}_N$. **ALL** combines both sharing schemes, using shared trunks for $Q$/policy and for the predictors simultaneously. On Ant and Fetch Reach, sharing only between predictors is mostly benign (with mild instability on Ant), whereas sharing the $Q$/policy trunk or combining both sharing schemes (**Q, POLICY** and **ALL**) significantly harms performance or leads to failed training. These results support our choice of fully separate networks in the main experiments.

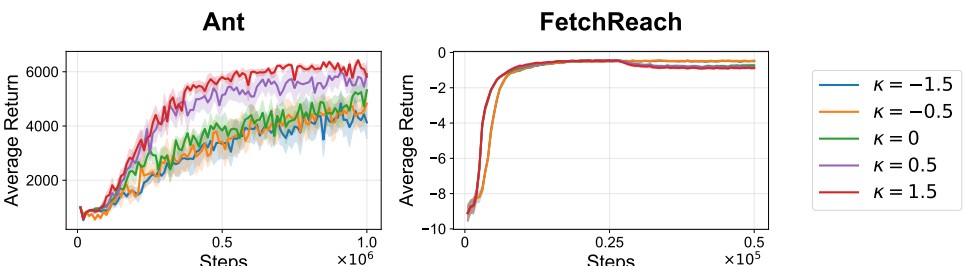

Figure 9: **Ablation study on $\kappa$ sensitivity.** We evaluated the sensitivity of our method to the choice of $\kappa$ in two environments. In our setup, $\lambda_\omega$ is trained using state–action pairs identified as outliers in the online disagreement distribution $d(s, a)$, with $\kappa$ controlling the $z$-score threshold for labeling symmetry-breaking samples. In *Ant*, performance remains stable and even improves as $\kappa$ increases ($\kappa > 0$), suggesting that symmetry breaking is relatively rare and only a small upper tail of $d(s, a)$ needs to be treated as breaking. In *Fetch Reach*, lower thresholds ($\kappa < 0$), which label a larger fraction of pairs as symmetry-breaking, yield better and more stable performance, while higher thresholds lead to late-stage degradation. Overall, the method is not overly sensitive to the exact value of $\kappa$; a coarse estimate of whether symmetry violations are sparse or common is sufficient to choose a robust threshold.

# D  EXPERIMENTAL DETAILS

In this section, we summarize baseline models, the group symmetries, environment details, and the hyperparameters used in each environment. The implemented group symmetries used in each environment are summarized in Table 1, and the corresponding group representations for each state and action space are summarized in Table 2, 3, 4, 5, 6. We summarize the common hyperparameters for DQN used in Grid-World, including those for PE-DQN in Table 7. For SAC, we use the default hyperparameters (Haarnoja et al., 2018), which are listed in Table 8, including those for PE-SAC. All the experiments were run on NVIDIA RTX 4090 GPUs.

Table 1: Symmetries of environments used in the experiments.

| Env | Implemented Symmetries |
|---|---|
| Grid-World | $C_4$ |
| Hopper | $\mathbb{Z}_2$ |
| Ant | $\mathbb{Z}_4$ |
| Swimmer | $\mathbb{Z}_2$ |
| Fetch | $SO(3)$ |
| UR5e | $SO(3)$ |

**Baselines.**  We evaluate all baselines under a unified experimental setup. For the Grid-World environment, the $Q$-network is built using the MDP-Homomorphic architecture (Van der Pol et al., 2020). For continuous-control tasks, every method employs the same EMLP backbone (Finzi et al., 2021b) to ensure architectural consistency.

**(1) Vanilla RL.** A standard non-equivariant MLP.

**(2) Exact-Equivariant.**  An exactly equivariant EMLP implemented following the RPP codebase (Finzi et al., 2021a), but *without* the GAN-style Adam $\beta$ hyperparameters introduced in Miyato et al. (2018) (we keep the default Adam betas).

**(3) RPP (Finzi et al., 2021a).**  We use the official RPP implementation with both value and policy RPP-based, as in the authors' main script. We do not use the non-equivariant value networks introduced in their ablations. We enable the GAN-style Adam $\beta$ settings for stability, mirroring their recommended configuration.

**(4) Approximately Equivariant RL (Park et al., 2025).**  We incorporate the relaxation mechanism of Park et al. (2025) into either an MDP-Homomorphic network (Van der Pol et al., 2020) or an EMLP backbone (Finzi et al., 2021b), rather than using the original escnn stack (Weiler & Cesa, 2019). This ensures that all baselines share an identical architecture and training pipeline. In practice, the DQN configuration closely follows the original design, while the SAC version applies learned per-channel scalars—initialized at 1—to the outputs of each equivariant linear layer. This constitutes a mechanism-level adaptation rather than a full reimplementation of their approach.

**Grid-World.**  The symmetry used in Grid-World is the Cyclic group $C_4$, as summarized in Table 1. It consists of a $15 \times 15$ grid, with observations given by the concatenated agent and goal positions $[x_{\text{agent}}, y_{\text{agent}}, x_{\text{goal}}, y_{\text{goal}}]$. The action space is $\{\uparrow, \leftarrow, \downarrow, \rightarrow\}$. Group representations are implemented as two concatenated 2D rotation matrices on the state space and a $4 \times 4$ permutation matrix on the action space. Rewards are defined as $+1$ for reaching the goal and $-0.01$ per step otherwise.

In additional Grid-World experiments, we consider two variants of this base environment. For the reward-level symmetry-breaking setting, we convert half of the (randomly selected) obstacles into passable cells that incur a reward penalty of $-0.5$ instead of blocking the agent. For the complex-dynamics variant, inspired by Gym's FrozenLake environment (Brockman et al., 2016), we introduce stochasticity in the action execution: when an action is issued, it is applied to the intended direction with probability $0.65$, and with probability $0.35$, the agent is moved to a different adjacent cell.

**Locomotion.** The symmetries used in each environment are summarized in Table 1. The state and action space representations, as well as the hyperparameters, are adopted from RPP (Finzi et al., 2021a). For Swimmer-v2, Finzi et al. (2021a) reports using the approximate symmetry $\mathbb{Z}_2 \times \mathbb{Z}_2$ (left-right, front-back symmetries), but the official code does not provide a correct implementation of this. Therefore, we instead use the exact symmetry $\mathbb{Z}_2$ (left-right symmetries) in our Swimmer experiments.

**Manipulation.** The symmetries used in each environment are summarized in Table 1. In Fetch Reach, the agent is trained to move the end-effector to a randomly sampled target position in each episode. The corresponding state and action spaces, together with their representations of the exploited symmetries, are provided in Table 5. A dense reward is given at every timestep as the negative Euclidean distance between the current end-effector and the goal position. In UR5e Reach, the agent is trained to reach a randomly sampled $SE(3)$ target pose in each episode. The corresponding state and action spaces with the representations of the exploited symmetries are provided in Table 6. A dense reward is given at every timestep as the negative weighted sum of the Euclidean distance (translational error) and the geodesic distance (rotational error) between the current end-effector and the goal poses. A weight of 0.19098621461 is applied to the geodesic distance term so that a $15°$ rotational error is treated as equivalent to a 0.05m translational error. We scale action of translation by 0.05 m for both tasks, and rotation by 0.2618 rad ($15°$) for UR5e Reach task.

**Overall.** The state and action representations used for the equivariant networks in each environment except Grid-World are shown in Table 2, 3, 4, 5, 6 (last column). In these tables, $V$ denotes an $n$-dimensional base representation, transformed by permutations for $\mathbb{Z}_n$ and by rotation matrices for $SO(3)$. $\mathbb{R}$ denotes a 1-dimensional scalar representation which is invariant under these group actions. $P$ denotes a 1-dimensional pseudoscalar representation, which is transformed by the sign of the permutation. (e.g., for Swimmer-v2, $P$ flips sign under left-right reflection of the body.) Note that powered representations such as $V^n$ indicate the direct sum of $n$ instances of the representation; this is given here as an example:

$$V^n = \bigoplus_{i=1}^{n} V.$$

Hyperparameters used for locomotion and manipulator (SAC) experiments are shown in Table 8. Those are shared across all tasks, unless specified in the table.

Table 2: Hopper-v2 state and action spaces with their representations

|  | Name | Description | Dim | Rep |
|---|---|---|---|---|
| State | Torso z | z-coordinate of the torso | 1 | $\mathbb{R}$ |
| | Orientation | Torso pitch angle | 1 | $P$ |
| | Thigh angle | Thigh joint angle | 1 | $P$ |
| | Leg angle | Leg joint angle | 1 | $P$ |
| | Foot angle | Foot joint angle | 1 | $P$ |
| | Torso velx | Linear velocity of torso (x) | 1 | $P$ |
| | Torso velz | Linear velocity of torso (z) | 1 | $\mathbb{R}$ |
| | Torso angvel | Angular velocity of torso (y) | 1 | $P$ |
| | Thigh angvel | Angular velocity of thigh hinge | 1 | $P$ |
| | Leg angvel | Angular velocity of leg hinge | 1 | $P$ |
| | Foot angvel | Angular velocity of foot hinge | 1 | $P$ |
| Action | Thigh | Torque applied on thigh joint | 1 | $P$ |
| | Leg | Torque applied on leg joint | 1 | $P$ |
| | Foot | Torque applied on foot joint | 1 | $P$ |

Table 3: Ant-v2 state and action spaces with their representations

|  | Name | Description | Dim | Rep |
|---|---|---|---|---|
| State | Torso z | z-coordinate of the torso | 1 | $\mathbb{R}$ |
|  | Torso quat | Orientation of the torso (quaternion) | 4 | $\mathbb{R}^4$ |
|  | Hip 1 angle | Angle between torso and front-left link | 1 |  |
|  | Hip 2 angle | Angle between torso and front-right link | 1 | $V$ |
|  | Hip 3 angle | Angle between torso and back-left link | 1 |  |
|  | Hip 4 angle | Angle between torso and back-right link | 1 |  |
|  | Ankle 1 angle | Angle between two front-left links | 1 |  |
|  | Ankle 2 angle | Angle between two front-right links | 1 | $V$ |
|  | Ankle 3 angle | Angle between two back-left links | 1 |  |
|  | Ankle 4 angle | Angle between two back-right links | 1 |  |
|  | Torso vel | Linear velocity of torso (x, y, z) | 3 | $\mathbb{R}^3$ |
|  | Torso angvel | Angular velocity of torso (x, y, z) | 3 | $\mathbb{R}^3$ |
|  | Hip 1 angvel | Angular velocity of front-left hip joint | 1 |  |
|  | Hip 2 angvel | Angular velocity of front-right hip joint | 1 | $V$ |
|  | Hip 3 angvel | Angular velocity of back-left hip joint | 1 |  |
|  | Hip 4 angvel | Angular velocity of back-right hip joint | 1 |  |
|  | Ankle 1 angvel | Angular velocity of front-left ankle joint | 1 |  |
|  | Ankle 2 angvel | Angular velocity of front-right ankle joint | 1 | $V$ |
|  | Ankle 3 angvel | Angular velocity of back-left ankle joint | 1 |  |
|  | Ankle 4 angvel | Angular velocity of back-right ankle joint | 1 |  |
| Action | Hip 1 | Torque on front-left hip joint | 1 |  |
|  | Hip 2 | Torque on front-right hip joint | 1 | $V$ |
|  | Hip 3 | Torque on back-left hip joint | 1 |  |
|  | Hip 4 | Torque on back-right hip joint | 1 |  |
|  | Ankle 1 | Torque on front-left ankle joint | 1 |  |
|  | Ankle 2 | Torque on front-right ankle joint | 1 | $V$ |
|  | Ankle 3 | Torque on back-left ankle joint | 1 |  |
|  | Ankle 4 | Torque on back-right ankle joint | 1 |  |

Table 4: Swimmer-v2 state and action spaces with their representations

|  | Name | Description | Dim | Rep |
|---|---|---|---|---|
| State | Orientation angle | Front tip angle | 1 | $P$ |
|  | Head joint angle | First rotor angle | 1 | $P$ |
|  | Tail joint angle | Second rotor angle | 1 | $P$ |
|  | x, y velocities | Tip velocities along x, y | 2 | $\mathbb{R}^2$ |
|  | Orientation angvel | Front tip angular velocity | 1 | $P$ |
|  | Head joint angvel | First rotor angular velocity | 1 | $P$ |
|  | Tail joint angvel | Second rotor angular velocity | 1 | $P$ |
| Action | Head joint | Torque on first rotor | 1 | $P$ |
|  | Tail joint | Torque on second rotor | 1 | $P$ |

Table 5: Fetch Reach state and action spaces with their representations

|        | Name        | Description                                     | Dim | Rep |
|--------|-------------|-------------------------------------------------|-----|-----|
| State  | EE pos      | End-effector position $(x, y, z)$               | 3   | $V$ |
|        | EE vel      | End-effector velocity $(v_x, v_y, v_z)$         | 3   | $V$ |
|        | Goal pos    | Goal position $(x, y, z)$                        | 3   | $V$ |
| Action | EE rel trans | Relative translation $(\Delta x, \Delta y, \Delta z)$ | 3 | $V$ |
|        | Gripper cmd | Gripper open/close control                      | 1   | $\mathbb{R}$ |

Table 6: UR5e Reach state and action spaces with their representations

|        | Name         | Description                                              | Dim | Rep   |
|--------|--------------|----------------------------------------------------------|-----|-------|
| State  | EE pos       | End-effector position $(x, y, z)$                        | 3   | $V$   |
|        | EE rot6d     | End-effector orientation (6D rep.)                       | 6   | $V^2$ |
|        | EE velp      | End-effector linear velocity $(v_x, v_y, v_z)$           | 3   | $V$   |
|        | EE velr      | End-effector angular velocity $(\omega_x, \omega_y, \omega_z)$ | 3 | $V$ |
|        | Goal pos     | Goal position $(x, y, z)$                                 | 3   | $V$   |
|        | Goal rot6d   | Goal orientation (6D rep.)                                | 6   | $V^2$ |
| Action | EE rel trans | Relative translation $(\Delta x, \Delta y, \Delta z)$    | 3   | $\mathbb{R}^3$ |
|        | EE rel rot   | Relative rotation (axis–angle) $(a_x, a_y, a_z)$          | 3   | $\mathbb{R}^3$ |

Table 7: Hyperparameters used in Grid-World (DQN) experiments.

| Hyperparameter | Value |
|----------------|-------|
| Optimizer | Adam (Kingma & Ba, 2015) |
| Learning rate | $3 \times 10^{-4}$ |
| Hidden size | $[256, 256]$ |
| Batch size | 256 |
| Discount factor $\gamma$ | 0.99 |
| Target network update rate $\tau$ | 0.005 |
| Replay buffer size | $1 \times 10^5$ |
| $\varepsilon$-greedy schedule | $1.0 \to 0.05$ ($5 \times 10^4$ steps) |
| $\lambda$, $\hat{P}_E$, $\hat{P}_N$ batch size | 256 |
| $\hat{P}_E$, $\hat{P}_N$ learning rate | $3 \times 10^{-4}$ |
| $\lambda$ learning rate | $1 \times 10^{-4}$ |
| #$\lambda$ warm-start steps | $2 \times 10^4$ ($\sim$30 obstacles), $4 \times 10^4$ (40 obstacles) |
| $\lambda$ prior bias | 0.5 |
| $\lambda$ hidden size | $[256, 256]$ |
| $\lambda$ gradient clipping | 1.0 |
| $\hat{P}_E$, $\hat{P}_N$ hidden size | $[256, 256]$ |
| $\hat{P}_E$, $\hat{P}_N$ gradient clipping | 1.0 |
| # $\hat{P}_E$, $\hat{P}_N$ gradient steps per update | 20 |
| Disagreement coefficient $\kappa$ (0 / 10 / 20 obstacles) | 1.5 |
| Quantile coefficient (30 / 40 obstacles) | 0.6 / 0.3 |
| # Threshold update interval steps | 200 |
| Threshold EMA $\beta$ | 0.2 |

Table 8: Hyperparameters used in locomotion and manipulation (SAC) experiments.

| Hyperparameter | Value |
|---|---|
| Optimizer | Adam (Kingma & Ba, 2015) |
| Actor learning rate | $3 \times 10^{-4}$ |
| Critic learning rate | $3 \times 10^{-4}$ |
| Temperature learning rate | $3 \times 10^{-4}$ |
| Entropy coefficient | auto-adjust (Haarnoja et al., 2018) |
| Batch size | 256 |
| Discount factor $\gamma$ | 0.99 |
| Target network update rate $\tau$ | 0.005 (0.004 for RPP Swimmer-v2) |
| Target entropy | $-0.5 \times \dim(\text{action})$ |
| Hidden size | $[256, 256]$ |
| Gradient clipping | 0.5 |
| $\lambda_\omega, \lambda_\zeta$ hidden size | $[128, 128]$ |
| $\hat{P}_E, \hat{P}_N$ hidden size | $[256, 256]$ |
| $\lambda_\omega, \lambda_\zeta, \hat{P}_E, \hat{P}_N$ batch size | 256 |
| # $\hat{P}_E, \hat{P}_N$ gradient steps | 2 |
| $\lambda_\omega, \lambda_\zeta$ learning rate | $1 \times 10^{-4}$ |
| $\lambda_\omega, \lambda_\zeta$ gradient clipping | 0.5 |
| $\lambda_\omega, \lambda_\zeta$ prior bias | 0.7685 |
| # Threshold update interval steps | 100 |
| Threshold EMA $\beta$ | 0.1 |
| Expectile regression coefficient $\tau_{\exp}$ | 0.8 |
| # Expectile action samples $M$ | 4 |

# E  EQUIVARIANCE ERROR AND ITS PROPAGATION UNDER SYMMETRY-BREAKING

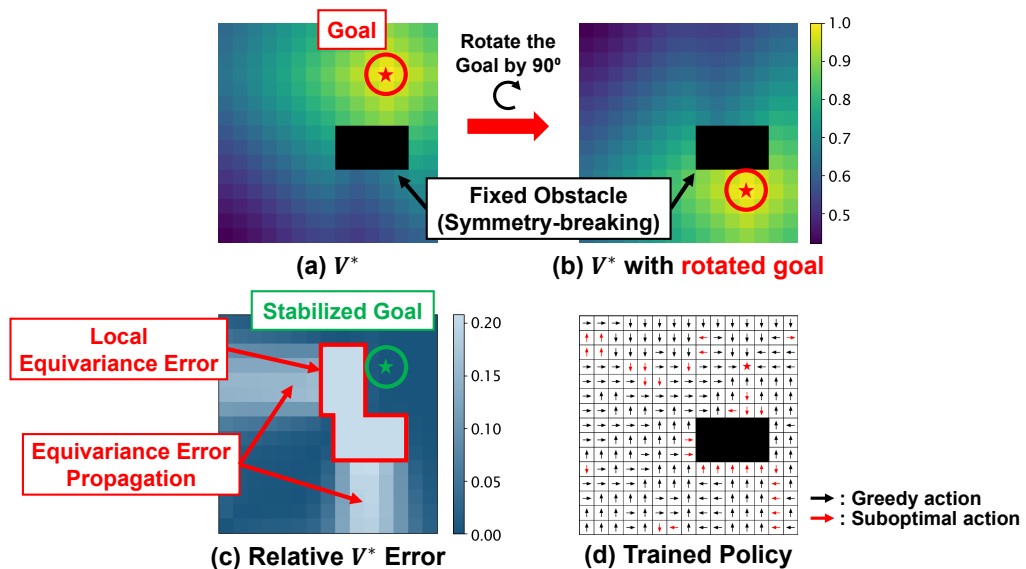

Figure 10: **Equivariance error under symmetry-breaking.** We assess rotational equivariance by comparing the base optimal value function $V^*$ with the value obtained after rotating the goal by $90°$ (red star) while keeping obstacles fixed (black cells), thereby breaking the symmetry. To intensify the effect of symmetry-breaking, we additionally incorporated stochasticity into the transition kernel in this figure. **(a)** Baseline optimal value $V^*$. **(b)** $V^*$ with the goal rotated by $90°$ while obstacles (black) remain fixed. **(c)** Per-state relative equivariance error $\left(\left|V^*(s) - V^*(gs)\right|/\left|V^*(s)\right|\right)$ with the goal stabilization. The sky-blue cells bordered by a red line coincide with the overlap between the original obstacle and its image under $g$, creating large local errors. The error then propagates outward, as reflected by the surrounding regions whose shading gradually darkens. This non-local propagation occurs for all $g \in G$ and has broader implications for equivariant RL training. **(d)** Greedy actions from an equivariant DQN. Red arrows denote suboptimal moves, illustrating that the learned policy inherits errors in symmetry-broken regions.

# F  THE USE OF LARGE LANGUAGE MODELS

In this paper, we used LLMs solely for text polishing and generating code snippets. Study design, theoretical results, algorithmic contributions, and all experiments/analyses were conceived and implemented by the authors. All code generated with LLM assistance was reviewed and verified by the authors.

