# OpenReview forum: "Partially Equivariant Reinforcement Learning in Symmetry-Breaking Environments"
_ICLR.cc/2026/Conference — ICLR 2026 Poster_

### Official Review · Reviewer_Gw6k · 2025-10-26

**Soundness:** 3
**Presentation:** 2
**Contribution:** 2
**Rating:** 4
**Confidence:** 4

**Summary:**

This paper introduced a novel partially equivariant reinforcement learning formulation and a novel algorithm to solve RL in imperfect equivariant MDP scenarios. The proposed partially equivariant reinforcement learning formulation, i.e., Partially group-Invariant MDP (PI-MDP), tackles symmetry-broken MDP by decomposition the MDP into an euqivariant MDP and a standard MDP. Later, authors proposed to identify symmetry breaking scenarios (state-action pairs) by dynamic-disagreements. Lastly, two algorithms, PE-DQN and PE-SAC are introduced to solve discrete and continuous PI-MDPs respectively. Experiments show the proposed method outperforms various baselines.

**Strengths:**

This paper Addressed partially invariant MDP by decomposing it into group-invariant MDP and a standard MDP. Then the two MDPs are merged by a measurable gating function. Beside theory, the paper introduced a forward disagreement measurement to practically estimate the gating. Experiments demonstrated the advantage of the proposed method.

**Weaknesses:**

Firstly, the paper do not explain well what is a symmetry breaking MDP. Authors gave an equation for group invariant MDP, could authors give equation for symmetry breaking MDP? Furthermore, Figure 1 showed an example of symmetry breaking MDP, I assume the obstacle is observable. Nevertheless, this example to me is more like an extrinsic equivariance [1], where the right subfigure is a new data, rather than a group-transformed seen observation (since the obstacle is not transformed). An equation of symmetry breaking MDP would clarify it.
The forward disagreement measurement seems relied on how good the unconstrained regressor can learn the forward dynamics. If the underlying MDP is group invariant, and the unconstrained regressor learns a bad dynamic, then this regressor will learn to large disagreement with the constrained regressor. In this case, the disagreement measurement fails to measure the equivariance of the MDP. Could authors explain more on this?

[1] A general theory of correct, incorrect, and extrinsic equivariance. D Wang, X Zhu, JY Park, M Jia, G Su, R Platt, R Walters - Advances in Neural Information Processing Systems, 2023

**Questions:**

Beside questions in weakness, I have following questions:
a. what is the alpha in equation9?
b. Could authors explain the detailed setup of the grid-world task? Especially how it break symmetry.

---

> ### Author Response · Authors · 2025-11-23
> **Official Comment by Authors**
>
> We appreciate the reviewer’s thoughtful engagement with our work. Below, we address each point in detail. We are grateful for the constructive perspectives and are open to additional input that may help further refine our study.
>
> &nbsp;
>
> ### **[Weakness 1: Definition of Symmetry Breaking MDP and Figure 1]**
> In the revised paper, we have formalized the definition of symmetry-breaking based on the reward and transition mismatches ($\epsilon_R$, $\epsilon_P$) introduced in Eq. (1) (see Definition 1). Specifically, we define an MDP as having **symmetry-breaking** at $(s,a)$ if $\epsilon_R(s,a) > 0$ or $\epsilon_P(s,a) > 0$. Conversely, symmetry is **preserved** only if these terms are zero.
>
> Regarding the distinction from **extrinsic equivariance (Figure 1)**: the reviewer is correct that if the obstacles were part of the input state and were transformed together, it would be a case of *extrinsic equivariance* [1]. However, in our grid-world setting, **obstacles are structural parameters fixed in the global frame that determine the transition dynamics**, but are **NOT** included in the agent’s state representation (which only contains agent/goal coordinates). Since the obstacle does not rotate when the agent rotates ($s \rightarrow gs$), the true transition dynamics $P_N$ violate the group-invariance condition: $P_N(gs’ | gs, ga) \neq P_N(s’|s,a)$. Consequently, from the agent’s perspective, this appears not as “new data” but as a **violation of the physical symmetry itself** within the transition dynamics. In the terminology of [1], this precisely corresponds to **incorrect equivariance**. We have revised the caption of Figure 1 to explicitly clarify that obstacles are not part of the state, preventing this ambiguity.
>
> [1] Wang et al. “A general theory of correct, incorrect, and extrinsic equivariance.”
>
> &nbsp;
>
> ### **[Weakness 2: Regarding the challenges in bad dynamics]**
>
> The reviewer raises a valid point: if the unconstrained regressor $\hat P_N$ fails to learn, $d(s,a)$ could be high due to prediction error rather than symmetry-breaking. We address this by relying on **relative disagreement detected via outlier filtering** (see Appendix B.1). In symmetric regions, both $\hat P_N$ and $\hat P_E$ converge towards the same target ($P_N = P_E$), making their difference random variance. In symmetry-breaking regions, however, $\hat P_E$ is **structurally incapable** of representing the true dynamics (averaging conflicting dynamics), whereas $\hat P_N$ **can** fit them. This creates a systematic divergence that dominates random errors.
>
> To validate this robustness, we added a stochastic GridWorld (40 obstacles) experiment (Figure 4b in the revised paper) where transitions include **stochastic slip noise**, making dynamics significantly harder to predict. This experiment serves as a rigorous test for the robustness of our disagreement-based gate even when the underlying regressors are imperfect.
>
> | Environment | $\\textbf{DQN}$ | $\\textbf{Equi-DQN}$ | $\\textbf{RPP-DQN}$ | $\\substack{\\textstyle\\textbf{ApproxEqui-} \\\\ \\textstyle\\textbf{DQN}}$ | $\\textbf{PE-DQN (Ours)}$ |
> | :--- | :---: | :---: | :---: | :---: | :---: |
> | **Stochastic, 40 obstacles** | $0.50 \\pm 0.09$ | $0.55 \\pm 0.08$ | $0.65 \\pm 0.05$ | $0.59 \\pm 0.07$ | $\\mathbf{0.69 \\pm 0.05}$ |
>
> Even in this high-entropy regime, our method successfully distinguishes symmetry-breaking regions via outlier filtering and significantly outperforms baselines (Figure 4b in the revised paper). This confirms that the **relative** disagreement signal via outlier filtering remains valid even with imperfect regressors.
>
> &nbsp;
>
> ### **[Question 1: What is the $\alpha$ in Eq.(9)?]**
> $\alpha$ refers to the temperature parameter used in the Soft Actor-Critic (SAC) algorithm [2]. This parameter balances the trade-off between entropy maximization and reward maximization. We have explicitly added this definition to the revised paper (near Eq. (9)) to ensure clarity.
>
> [2] Haarnoja et al. “Soft actor-critic algorithms and applications”
>
> &nbsp;
>
> ### **[Question 2:Detailed setup of the grid-world task]**
> We provided the detailed grid-world setup in Experiments and Appendix D. The state consists of the coordinates of the agent and the goal: $s = [ x_{agent}, y_{agent}, x_{goal}, y_{goal} ]$. Crucially, the obstacles are **not** part of the state. We have added these explanations to the Experiments section of the revised manuscript.
>
> In this setup, symmetry is broken because the fixed layout of obstacles blocks movement in specific cells. For example, an action “up” might be valid at $s$, but after rotating $90^{\circ}$, the corresponding “right” at $gs$ might be blocked by a fixed obstacle. Consequently, from the agent’s perspective, this appears not as “new data” but as a violation of the physical symmetry itself within the transition dynamics.

---

### Official Review · Reviewer_tm3G · 2025-10-31

**Soundness:** 3
**Presentation:** 3
**Contribution:** 2
**Rating:** 4
**Confidence:** 3

**Summary:**

This paper tackles the problem of symmetry-breaking in equivariant reinforcement learning, where standard equivariant methods fail when environment symmetries are not held. The authors propose a novel framework called the Partially group-Invariant MDP (PI-MDP) and practical algorithms (PE-DQN, PE-SAC) that use a learned gating function to dynamically switch between an equivariant and a standard network, applying equivariance only where symmetries are satisfied. Experiment results demonstrate that the method successfully combines the sample efficiency of equivariance with robustness to symmetry-breaking, outperforming the baselines.

**Strengths:**

1. The paper is generally well-written. The problem setup is clear, and the main idea is easy to follow.
2.  The core idea of using a local and learned gating mechanism to handle symmetry-breaking is well-motivated.
2. The paper provides a theoretical analysis of the error propagation from local symmetry-breaking.

**Weaknesses:**

1. The method introduces significant additional complexity: training two dynamics models, with corresponding two policy/value networks and two gating functions. The paper does not conduct the ablation analysis on these components, which would strengthen the impact.

2. The current approach relies on dynamic disagreement to detect symmetry-breaking. This might be less effective for environments where symmetry is broken primarily in the reward function rather than the dynamics. The paper notes this limitation but does not explore alternatives or the performance impact in such scenarios.

3. In lines 262~263, the author states that " We assume those symmetry-breaking disagreements as outliers in the online distribution of $d(s, a)$. We label outliers with $y(s, a) ∈ {0, 1}$ using an online detector." Such a core assumption that its outliers reliably indicate symmetry violations is potentially fragile. For example, in stochastic environments, high variance in transitions could inflate $d(s, a)$ values even in symmetric regions, while in cases where the dynamics models are also inaccurate, leading to an unreliable gating function.

**Questions:**

1. Was an ablation or sensitivity analysis performed on the parameters used in the proposed method?
2. The paper uses a hard gating strategy for simplicity. Did the authors experiment with a soft, probabilistic gating mechanism?

---

> ### Author Response · Authors · 2025-11-23
> **Official Comment by Authors (1/3)**
>
> We appreciate the reviewer’s thoughtful engagement with our work. Below, we address each point in detail. We are grateful for the constructive perspectives and are open to additional input that may help further refine our study.
>
> &nbsp;
>
> ### **[Weakness 1: Ablation on additional complexity]**
>
> We acknowledge the complexity concern. To justify our design choices, we conducted ablation studies on (1) unifying gating functions and (2) sharing network trunks. Please refer to Appendix C for the detailed learning curves.
>
>
> **(1)  Unifying gating functions ($\lambda_{\zeta} \approx \max_{a} \lambda_{\omega}$)**: We compared our state-gate $\lambda_\zeta (s)$ against a heuristic that estimates $\max_a \lambda_\omega (s,a)$ by sampling $K$ actions without training additional $\lambda_\zeta (s)$.
>
> | Environment | $\\substack{\\textstyle\\textbf{Using separate} \\\\ \\textstyle\\textbf{gate}}$ | $\\substack{\\textstyle\\textbf{Using } \\max_a \\textbf{ from} \\\\ \\textstyle\\textbf{4 sampled actions}}$ | $\\substack{\\textstyle\\textbf{Using } \\max_a \\textbf{ from} \\\\ \\textstyle\\textbf{8 sampled actions}}$ |
> | :--- | :---: | :---: | :---: |
> | **Ant** | $\\mathbf{6137.39 \\pm 95.20}$ | $6059.06 \\pm 229.50$ | $6125.72 \\pm 454.45$ |
> | **Fetch Reach** | $\\mathbf{-0.48 \\pm 0.03}$ | $-0.64 \\pm 0.06$ | $-0.56 \\pm 0.06$ |
>
> The results indicate that the Sampled-max heuristic is broadly competitive, but in Ant it shows less stable learning than the trainable state gate. As shown in Figure 6 (Appendix C), the learned gate produces smoother training curves, especially when symmetry breaking is sparse and a small action sample may fail to hit the informative regions.
>
>
> **(2)  Trunk sharing**: We tested sharing equivariant trunks for critic/actor (Q/$\pi$) or dynamics.
>
> | Environment | $\\substack{\\textstyle\\textbf{Separate} \\\\ \\textstyle\\textbf{(Original)}}$ | $\\substack{\\textstyle Q/\\pi \\textbf{ Trunk} \\\\ \\textstyle\\textbf{Sharing}}$ | $\\substack{\\textstyle\\textbf{Dynamics Trunk} \\\\ \\textstyle\\textbf{Sharing}}$ | $\\substack{\\textstyle\\textbf{All Trunk} \\\\ \\textstyle\\textbf{Sharing}}$ |
> | :--- | :---: | :---: | :---: | :---: |
> | **Ant** | $\\mathbf{6137.39 \\pm 95.20}$ | $-185.43 \\pm 115.48$ | $5631.10 \\pm 391.87$ | $94.07 \\pm 93.28$ |
> | **Fetch Reach** | $\\mathbf{-0.48 \\pm 0.03}$ | $-1.21 \\pm 0.04$ | $\\mathbf{-0.48 \\pm 0.03}$ | $-1.59 \\pm 0.22$ |
>
> The results show that sharing dynamics trunk is feasible (supervised learning), but sharing $Q/\pi$ trunks is harmful (RL bootstrapping). Since $Q_E$ and $Q_N$ approximate fundamentally different value functions, sharing a trunk causes feature conflict, which is amplified by bootstrapping. Thus, we maintain separate networks for stability.
>
> &nbsp;
>
> ### **[Weakness 2: Can our approach address symmetry-breaking primarily in the reward function?]**
> We appreciate the reviewer’s suggestion. To handle environments where rewards break symmetry, we extended our framework by adding a reward head to the environment model predictors (predictors in short) and including reward distance in the disagreement score $d(s,a)$.
>
>
> | Environment | $\\textbf{DQN}$ | $\\textbf{Equi-DQN}$ | $\\textbf{RPP-DQN}$ | $\\substack{\\textstyle\\textbf{ApproxEqui-} \\\\ \\textstyle\\textbf{DQN}}$ | $\\textbf{PE-DQN (Ours)}$ |
> | :--- | :---: | :---: | :---: | :---: | :---: |
> | **GridWorld, 10 obstacles** | $0.52 \\pm 0.07$ | $0.59 \\pm 0.07$ | $0.84 \\pm 0.02$ | $0.34 \\pm 0.10$ | $\\mathbf{0.90 \\pm 0.01}$ |
> | **GridWorld, 30 obstacles** | $-0.02 \\pm 0.08$ | $-0.01 \\pm 0.18$ | $0.19 \\pm 0.11$ | $-0.07 \\pm 0.20$ | $\\mathbf{0.58 \\pm 0.10}$ |
>
> We evaluated this on GridWorld variants where **half of the obstacles are passable with a reward penalty** while the others remain blocking. This creates a challenging scenario with **mixed transition and reward asymmetries**. As shown above, our extended method significantly outperforms baselines, demonstrating its capability to detect and handle both types of symmetry-breaking simultaneously (see Figure 4a of the revised paper).

---

> ### Author Response · Authors · 2025-11-23
> **Official Comment by Authors (2/3)**
>
> ### **[Weakness 3: Limitations in Defining Symmetry-Breaking Regions]**
>
> Please note that our method relies on **relative difference** in distributional errors rather than absolute values. In our newly added stochastic experiments (Figure 4b of the revised paper and the table below), we explicitly measure the divergence between predicted transition distributions. Since we focus on **relative** discrepancies, regions with **significantly larger divergences** (caused by symmetry violations) remain distinguishable from the baseline noise caused by stochastic dynamics via outlier filtering (see Appendix B.1. Disagreement thresholding and label generation part). Specifically, by dynamically labeling the top $\alpha$-percentile (or z-score outliers) of the disagreement distribution, our framework effectively normalizes global variance shifts caused by stochasticity.
>
> To empirically test this robustness, we evaluated a **stochastic** GridWorld (40 obstacles) where transitions include stochastic slip noise, making dynamics significantly harder to predict. As shown in the table below (and Figure 4b in the revised paper), our method significantly outperforms baselines even in the high-variance regime. This confirms that our adaptive gating mechanism remains robust in practice.
>
> | Environment | $\\textbf{DQN}$ | $\\textbf{Equi-DQN}$ | $\\textbf{RPP-DQN}$ | $\\substack{\\textstyle\\textbf{ApproxEqui-} \\\\ \\textstyle\\textbf{DQN}}$ | $\\textbf{PE-DQN (Ours)}$ |
> | :--- | :---: | :---: | :---: | :---: | :---: |
> | **Stochastic, 40 obstacles** | $0.50 \\pm 0.09$ | $0.55 \\pm 0.08$ | $0.65 \\pm 0.05$ | $0.59 \\pm 0.07$ | $\\mathbf{0.69 \\pm 0.05}$ |
>
> &nbsp;
>
> ### **[Question 1: Ablation or sensitivity analysis on hyperparameters]**
>
> | Environment | $\\substack{\\textstyle\\kappa = -1.5 \\\\ \\textstyle\\text{(uppertail 93\\%)}}$ | $\\substack{\\textstyle\\kappa = -0.5 \\\\ \\textstyle\\text{(uppertail 69\\%)}}$ | $\\substack{\\textstyle\\kappa = 0.0 \\\\ \\textstyle\\text{(uppertail 50\\%)}}$ | $\\substack{\\textstyle\\kappa = 0.5 \\\\ \\textstyle\\text{(uppertail 31\\%)}}$ | $\\substack{\\textstyle\\kappa = 1.5 \\\\ \\textstyle\\text{(uppertail 7\\%)}}$ |
> | :--- | :---: | :---: | :---: | :---: | :---: |
> | **Ant** | $4328.05 \\pm 415.66$ | $4539.14 \\pm 433.85$ | $5143.05 \\pm 385.26$ | $5690.20 \\pm 489.01$ | $\\mathbf{6137.39 \\pm 95.20}$ |
> | **Fetch Reach** | $-0.49 \\pm 0.03$ | $\\mathbf{-0.47 \\pm 0.04}$ | $-0.73 \\pm 0.07$ | $-0.81 \\pm 0.06$ | $-0.88 \\pm 0.05$ |
>
> Per the reviewer’s request, we conducted additional experiments on the sensitivity to the threshold parameter $\kappa$ (z-score for outlier detection) (Figure 9 of the revised paper). The results show that the method is not overly sensitive to the exact value of $\kappa$ and reasonably robust to a range of $\kappa$, provided that the parameter broadly reflects the prevalence of symmetry-breaking (rare vs. common). In Ant (where symmetry breaking is rare), performance remains stable and high across a wide range of higher thresholds ($\kappa > 0$). In FetchReach (where symmetry breaking is frequent), lower thresholds ($\kappa < 0$) consistently yield better results. This confirms that exhaustive hyperparameter search is not necessary; a rough estimate of whether symmetry violations are sparse or dense is sufficient to select a robust $\kappa$.

---

> ### Author Response · Authors · 2025-11-23
> **Official Comment by Authors (3/3)**
>
> ### **[Question 2: Ablation on soft gating mechanism]**
> As suggested by the reviewer, we conducted additional experiments comparing **hard gating (ours)**, **soft gating**, and **fixed soft gating ($\lambda=0.5$)**. We included the fixed $\lambda$ setting to mimic RPP-style residual mixing without the instability of a learnable gate.
>
> | Environment | $\\substack{\\textstyle\\textbf{Hard Gating} \\\\ \\textstyle(\\lambda \\in \\{0,1\\})}$ | $\\substack{\\textstyle\\textbf{Soft Gating} \\\\ \\textstyle\\text{(Learnable } \\lambda \\in [0,1])}$ | $\\substack{\\textstyle\\textbf{Soft Gating} \\\\ \\textstyle\\text{(Fixed } \\lambda = 0.5)}$ | $\\textbf{RPP}$ |
> | :--- | :---: | :---: | :---: | :---: |
> | **Ant (500K steps)** | $\\mathbf{5680.82 \\pm 138.78}$ | $4261.72 \\pm 421.00$ | $5212.06 \\pm 199.18$ | $4004.06 \\pm 224.70$ |
> | **Ant (1M steps)** | $\\mathbf{6137.39 \\pm 95.20}$ | $5706.71 \\pm 436.53$ | $5370.14 \\pm 464.87$ | $6045.45 \\pm 129.27$ |
> | **Fetch Reach (25K)** | $\\mathbf{-0.50 \\pm 0.03}$ | $-5.56 \\pm 1.05$ | $-0.57 \\pm 0.04$ | $-0.75 \\pm 0.03$ |
> | **Fetch Reach (50K)** | $-0.48 \\pm 0.03$ | $-5.66 \\pm 1.72$ | $\\mathbf{-0.47 \\pm 0.03}$ | $-0.63 \\pm 0.03$ |
>
> The results show that **soft gating generally leads to unstable training**, particularly in FetchReach. We attribute this to “gradient isolation”: $\lambda$ is trained via a separate disagreement objective without feedback from the critic training. Consequently, fluctuations in $\lambda$ create a “moving target” for the value function, destabilizing learning.
>
> We observed that simply fixing $\lambda=0.5$ avoids this catastrophic failure. This confirms that the primary source of instability in soft gating is the non-stationary nature of the learned gate, rather than the soft blending itself. Given this finding, we adopt **hard gating** as it acts as a discrete router, selecting a single expert ($Q_E$ or $Q_N$). This eliminates the ambiguity, providing the most consistent stability and efficiency across tasks. For detailed learning curves, please refer to Figure 7 in Appendix C of the revised paper.

---

> ### Comment · Reviewer_tm3G · 2025-11-26
>
> Thanks for the authors' detailed reply. I have no further questions and I'm willing to raise my score. I suggest the authors to add these additional experiment results and discussions into the final manuscript which will further improves the overall quality of the apper.

---

> ### Author Response · Authors · 2025-11-28
> **Official Comment by Authors**
>
> We really appreciate the reviewer's careful evaluation and the positive assessment. We have included all additional experiments and discussions as suggested to further improve the paper.

---

### Official Review · Reviewer_8FGx · 2025-11-01

**Soundness:** 4
**Presentation:** 4
**Contribution:** 3
**Rating:** 8
**Confidence:** 4

**Summary:**

This paper proposes incorporating equivariance with reinforcement learning only for certain state-action pairs. The authors propose a partially group-invariant MDP that uses a gating mechanism to switch between a group-invariant or a standard MDP. They bound the optimal Q value so that errors are not propagated when the gating function routes correctly. Experiments on a gridworld domain and continuous control and robotics domains show improved performance over baselines.

Overall, this paper proposes a good way to handle symmetry violations in RL with a gating mechanism, such that one does not have to resort to global notions of exact or approximate equivariance and can use local definitions.

**Strengths:**

- The authors consider an important problem in equivariance and RL, specifically that symmetry violations are often local in practice. The proposed gating mechanism seems like a good and straightforward approach.
- The PI-MDP formulation is well-motivated and the authors also provide theoretical analysis on the error of the optimal Q function, given the optimal gating.
- The use of disagreement to discover symmetry violations seems like a good choice.
- Experiments are carried out on various domains and the results show that the PI-MDP framework often outperforms exactly equivariant RL.

**Weaknesses:**

See questions.

**Questions:**

- The theory is assumed for strict binary gating, while the experiments seem to use a soft gating mechanism for $\lambda_\omega$. Did you find that soft gating works better empirically over strict gating?
- I didn't fully understand why a separate state-only gating function was needed. Can you not just use the Q-gating function $\lambda_\omega$ for the policy? Is this only necessary for continuous actions? I feel like making $\lambda_\zeta$ learnable can potentially lead to mismatch between $\lambda_\omega$ and $\lambda_\zeta$.
- One weakness I can see is that since the PI-MDP framework requires separate critics and policies, using separate networks can require more samples than with a single MDP. For example, if the gating is routed to $M_E$, then only the $Q_E, \pi_E$ networks are optimized and vice versa, leading to lower sample efficiency. This is perhaps why a soft gating mechanism is preferable as well.
- Another question is that if we assume that symmetry violations in certain state-action pairs, then there are large regions where $M_E$ and $M_N$ coincide and thus their respective $Q$ and $\pi$ are the same/similar. Using separate networks can be somewhat parameter inefficient in this case. Have you tried sharing portions of the networks (such as a common trunk) among the critics and policies, and maybe also for the dynamics models?

---

> ### Author Response · Authors · 2025-11-23
> **Official Comment by Authors (1/2)**
>
> We appreciate the reviewer’s thoughtful engagement with our work. Below, we address each point in detail. We are grateful for the constructive perspectives and are open to additional input that may help further refine our study.
>
> &nbsp;
> ### **[Question 1: About the gating mechanism in the paper]**
>
> First, we clarify that we adopt **hard gating** (via Bernoulli sampling of the continuous probability $\lambda_{\omega}$ and $\lambda_{\zeta}$) in all main experiments, consistent with the theoretical derivation in Section 5 and Remark 1.
>
> Still, to address the reviewer’s question regarding the empirical comparison, we conducted additional ablation experiments comparing **hard gating**, **soft gating**, and **fixed soft gating ($\lambda=0.5$, mimicking RPP [1])**.
>
> | Environment | $\\substack{\\textstyle\\textbf{Hard Gating} \\\\ \\textstyle(\\lambda \\in \\{0,1\\})}$ | $\\substack{\\textstyle\\textbf{Soft Gating} \\\\ \\textstyle\\text{(Learnable } \\lambda \\in [0,1])}$ | $\\substack{\\textstyle\\textbf{Soft Gating} \\\\ \\textstyle\\text{(Fixed } \\lambda = 0.5)}$ | $\\textbf{RPP}$ |
> | :--- | :---: | :---: | :---: | :---: |
> | **Ant (500K steps)** | $\\mathbf{5680.82 \\pm 138.78}$ | $4261.72 \\pm 421.00$ | $5212.06 \\pm 199.18$ | $4004.06 \\pm 224.70$ |
> | **Ant (1M steps)** | $\\mathbf{6137.39 \\pm 95.20}$ | $5706.71 \\pm 436.53$ | $5370.14 \\pm 464.87$ | $6045.45 \\pm 129.27$ |
> | **Fetch Reach (25K)** | $\\mathbf{-0.50 \\pm 0.03}$ | $-5.56 \\pm 1.05$ | $-0.57 \\pm 0.04$ | $-0.75 \\pm 0.03$ |
> | **Fetch Reach (50K)** | $-0.48 \\pm 0.03$ | $-5.66 \\pm 1.72$ | $\\mathbf{-0.47 \\pm 0.03}$ | $-0.63 \\pm 0.03$ |
>
> The results show that **soft gating generally leads to instability** and slower convergence. We attribute this to “gradient isolation”: since $\lambda_\omega$ is trained via a separate objective without critic feedback, its fluctuations under soft gating create non-stationary noise in the Bellman target. In contrast, hard gating acts as a discrete router, selecting the single most reliable expert immediately. This yields cleaner learning signals, supported by the superior early performance in Ant (at 500K steps). For learning curves, please refer to Figure 7 in Appendix C of the revised paper.
>
> [1] Finzi et al. Residual Pathway Priors for Soft Equivariance Constraints (2021)
>
> &nbsp;
> ### **[Question 2: Can we directly use the Q-gating function for the policy?]**
> Although a unified gating function is desirable, using an action-dependent gate $\lambda_{\omega} (s,a)$ in the policy introduces an **intractable normalizing constant**, which breaks the reparameterization trick required for SAC (see Appendix A.4). Thus, we use a state-dependent gate $\lambda_{\zeta} (s)$ trained to approximate the upper tail of the symmetry-breaking signal ($\max_{a} \lambda_{\omega} (s,a)$).
>
> To address the reviewer's concern about potential mismatch, we compared our method against a simpler **Sampled-max** heuristic, where we estimate $\max_{a} \lambda_{\omega} (s,a)$ by taking the maximum over $K$ sampled actions instead of learning a separate network.
>
> | Environment | $\\substack{\\textstyle\\textbf{Using separate} \\\\ \\textstyle\\textbf{gate}}$ | $\\substack{\\textstyle\\textbf{Using } \\max_a \\textbf{ from} \\\\ \\textstyle\\textbf{4 sampled actions}}$ | $\\substack{\\textstyle\\textbf{Using } \\max_a \\textbf{ from} \\\\ \\textstyle\\textbf{8 sampled actions}}$ |
> | :--- | :---: | :---: | :---: |
> | **Ant** | $\\mathbf{6137.39 \\pm 95.20}$ | $6059.06 \\pm 229.50$ | $6125.72 \\pm 454.45$ |
> | **Fetch Reach** | $\\mathbf{-0.48 \\pm 0.03}$ | $-0.64 \\pm 0.06$ | $-0.56 \\pm 0.06$ |
>
> The results show that the Sampled-max heuristic is broadly competitive, achieving similar final performance in both settings. However, in Ant, it tends to exhibit slightly less stable learning curves than the trainable state gate. As illustrated in Figure 6 (Appendix C of the revised paper), the learned gate provides smoother and more reliable training, especially when symmetry-breaking is sparse and may be missed by a small set of sampled actions.

---

> ### Author Response · Authors · 2025-11-23
> **Official Comment by Authors (2/2)**
>
> ### **[Question 3: Concern about the inefficiency introduced by separate networks]**
> We understand the reviewer's concern that separate networks might imply sample inefficiency. However, our method empirically outperforms baselines in sample efficiency (Figures 3 and 5). We attribute this to two main factors that outweigh the cost of separation:
>
> 1. **Exploiting symmetry**: By routing samples to the equivariant expert ($Q_E$) in symmetric regions, the agent exploits the strong inductive bias of the symmetry. This allows $Q_E$ to learn optimal policies with significantly fewer samples than a standard network, effectively acting as a **highly efficient learner** that accelerates the overall training progress.
> 2. **Data sharing & knowledge transfer**: Since both networks are trained off-policy using a **shared replay buffer**, **no experience is wasted**. Furthermore, the rapidly learning $Q_E$ provides a high-quality TD target for $Q_N$ (when the next state is symmetric), creating a transfer mechanism that accelerates $Q_N$'s convergence even with fewer direct updates.
>
> Thus, our framework effectively combines the rapid learning of equivariance with the asymptotic robustness of the unconstrained network, resulting in superior overall efficiency.
>
> &nbsp;
>
> ### **[Question 4: Using the sharing portion of the networks]**
> We appreciate the reviewer’s suggestion to improve parameter efficiency. To evaluate this, we conducted additional ablation studies comparing our original fully separate architecture against variants with shared equivariant trunks for the critic/policies, dynamics, or all components.
>
> | Environment | $\\substack{\\textstyle\\textbf{Separate} \\\\ \\textstyle\\textbf{(Original)}}$ | $\\substack{\\textstyle Q/\\pi \\textbf{ Trunk} \\\\ \\textstyle\\textbf{Sharing}}$ | $\\substack{\\textstyle\\textbf{Dynamics Trunk} \\\\ \\textstyle\\textbf{Sharing}}$ | $\\substack{\\textstyle\\textbf{All Trunk} \\\\ \\textstyle\\textbf{Sharing}}$ |
> | :--- | :---: | :---: | :---: | :---: |
> | **Ant** | $\\mathbf{6137.39 \\pm 95.20}$ | $-185.43 \\pm 115.48$ | $5631.10 \\pm 391.87$ | $94.07 \\pm 93.28$ |
> | **Fetch Reach** | $\\mathbf{-0.48 \\pm 0.03}$ | $-1.21 \\pm 0.04$ | $\\mathbf{-0.48 \\pm 0.03}$ | $-1.59 \\pm 0.22$ |
>
> The results show a clear difference between the choices. This stems from the fundamental difference between supervised learning and reinforcement learning.
>
> Dynamics trunk sharing was **feasible** (supervised regression) **but less stable than separating them**. Unlike Q-sharing (described below), this approach avoids total failure because both heads regress towards the same ground-truth transitions. However, the shared trunk must simultaneously satisfy conflicting objectives: the strict equivariance of $P_E$ and the unconstrained fitting of $P_N$. This introduces optimization challenges, leading to instability in complex tasks (see orange curve in Figure 8 of the revised paper).
>
> In contrast, critic/policy trunk sharing was **harmful** (RL bootstrapping). RL agents rely on bootstrapping, where targets depend on the network itself. Since $Q_E$ and $Q_N$ aim to approximate fundamentally different value functions (group-averaged vs exact), they impose incompatible requirements on the shared features. Forcing a single trunk to satisfy both leads to feature interference, which is then amplified by the bootstrapping loop, destabilizing the entire learning process.
>
> Therefore, while sharing dynamics trunks is possible, it can hurt stability. To ensure the highest robustness and sample efficiency, our PI-MDP framework adopts a fully separate architecture, preventing any negative transfer between components. For detailed learning curves, please refer to Figure 8 in Appendix C of the revised paper.

---

### Author Response · Authors · 2025-12-02
**Summary of Revisions and Response to Reviewers**

We thank the Area Chair for the time and careful consideration. Below is a clear summary of our main contributions and how we have addressed all reviewer comments in the rebuttal and revised paper.

&nbsp;

### **Executive Summary**

This work studies why strictly equivariant RL becomes unreliable when real-world symmetries are only partially valid. We show theoretically that local symmetry violations cause Bellman backup errors that accumulate and corrupt value functions globally.
To address this, we introduce **Partially Group-Invariant MDPs (PI-MDPs)** and a practical gating mechanism that selectively applies symmetry only when it holds.

The revised paper now adds new experiments including extensive ablation studies and formal clarifications, which we believe directly address all reviewers’ concerns. **Crucially, one reviewer increased their score after the rebuttal (Reviewer tm3G: 4 → 6)**, confirming that the rebuttal and revisions successfully resolved the reviewer’s concern.

&nbsp;

### **Core Contributions**
  **1. Problem**

Existing equivariant RL relies on the assumption that rewards and transition dynamics are invariant under group transformations. In practice, this assumption is often violated (e.g., joint limits, singularities). We theoretically show that even small local violations propagate through group-invariant Bellman backups, leading to globally corrupted Q-values.

**2. Solution**

We introduce Partially Group-Invariant MDPs (PI-MDPs) and a practical gating mechanism that selectively applies symmetry only where it holds, identifying symmetric regions using disagreement between symmetry and non-symmetric dynamics models.

**3. Evidence**

- **Theory**: The fixed point under PI-MDP matches the true MDP’s fixed point when the gate is correct
- **Experiments**:  Across discrete task (with controllable symmetry-breaking), continuous locomotion, and robotic manipulation tasks, our approach outperforms all baselines, including approximate equivariant methods, in both sample efficiency and final performance

&nbsp;

### **Revised Parts in the Updated Paper**

**1. Stochastic and reward robustness**: We extended dynamics models and added new experiments to evaluate robustness to reward-level symmetry breaking and stochastic dynamics (Figure 4). Our method remains consistently robust and outperforming (Reviewer tm3G, Reviewer Gw6k).

**2. Ablation - Policy gating (Appendix C)**: We compared the learnable policy gate to a sampled-max heuristic. While the performance is comparable, the learnable gate is preferred for its robustness against sparse symmetry breaking (Figure 6) (Reviewer 8FGx, Reviewer tm3G).

**3. Ablation - Gating mechanism (Appendix C)**: Soft blends of equivariant and non-equivariant networks were tested but showed instability. Hard gating is confirmed to be more reliable, which well-aligns with the theory (Figure 7) (Reviewer 8FGx, Reviewer tm3G).

**4. Ablation - Shared trunks (Appendix C)**: Sharing network trunks reduced parameter count but caused instability, supporting our decision to use separate networks (Figure 8) (Reviewer 8FGx, Reviewer tm3G).

**5. Ablation - Threshold sensitivity (Appendix C)**: We evaluated the sensitivity of the disagreement threshold and found PI-MDP robust to a wide range of reasonable values (Figure 9) (Reviewer tm3G).

**6. Clarifying the definition of symmetry-breaking**: We added Definition 1 to clearly formalize what “symmetry-breaking” means in the context of MDPs and aligned terminology with prior work as suggested by reviewers (Reviewer Gw6k).

&nbsp;

### **Mapping Reviewer Concerns to Revisions**
| Reviewer | Key Concerns | How the Revision Addresses Them |
|:--------:|--------------|---------------------------------|
| **8FGx** | Gating mechanism; policy gating; inefficiency of separate networks; network sharing | **Resolved**: Ablations 1–3 in Appendix C address gating, alternatives, and trunk sharing |
| **tm3G** | Too many networks; reward-related symmetry-breaking; inaccurate dynamics; sensitivity; need soft-gating comparison | **Resolved**: Added new experiments on reward-breaking & stochastic dynamics, Ablations 1–4 in Appendix C; reviewer increased score 4 → 6 |
| **Gw6k** | Ambiguous symmetry-breaking definition, unclear Figure 1; difficulty with bad dynamics; unclear notations & env setup | **Resolved**: Added Definition 1; clarified terminology; expanded experiment details; added experiments with complex dynamics |

---

### Meta-Review · Area_Chair_nfMZ · 2026-01-07

**Summary:**

1. The theory uses binary gating, but the experiments use soft gating. *8FGx*
2. The PI-MDP framework requires separate critics and actors making it potentially less sample efficient.  It also potentially parameter inefficient *8FGx*
3. the method is complicated and needs ablations to justify the complexity.  *tm3G*
4. the paper should explore more the setting where only the reward function breaks symmetry. *tm3G*
5. the assumption that symmetry-breaking state-action pairs are detectable as outliers is shaky, especially for stochastic environments. *tm3G*
6. the definition and explanation of symmetry-breaking MDP is unclear *Gw6k*
7. failure to learn forward dynamics may result in false detection of symmetry breaking *Gw6k*

**Reviewer Concerns:**

1. The authors clarified hard gating was used in experiments and added experiments with soft gating.
2. The authors point to concrete results showing their method is sample efficient.  They perform additional design experiments that show increased parameter sharing comes at a stability/performance cost.
3. The authors added ablations on different aspects of their method.
4. The authors modified their method to handle cases where only the reward function breaks symmetry.
5. The authors address this concern by adding an experiment where symmetry breaking is stochastic and show their method is robust.
6. The authors clarified the definition of symmetry breaking in MDPs.
7. The authors explain why their method should be robust to errors and point to their new stochastic experiment to support it.

**Reviewer Scores:**

- *8FGx* gave the paper an 8.  Their concerns were well addressed and they may have considered a 10.
- *tm3G* raised their score from 4 to 6.
- *Gw6k* gave a 4, but their concerns were addressed and there is a good chance of raising their score to a 6.

---

### Decision · Program_Chairs · 2026-01-26

Accept (Poster)